# Quantitative analysis of printed nanostructured networks using high-resolution 3D FIB-SEM nanotomography

Cian Gabbett [1,3], Luke Doolan[1,3], Kevin Synnatschke [1], Laura Gambini [1], Emmet Coleman [1], Adam G. Kelly[1], Shixin Liu[1], Eoin Caffrey [1], Jose Munuera [1,2], Catriona Murphy[1], Stefano Sanvito [1], Lewys Jones [1] & Jonathan N. Coleman [1] ✉

Networks of solution-processed nanomaterials are becoming increasingly important across applications in electronics, sensing and energy storage/generation. Although the physical properties of these devices are often completely dominated by network morphology, the network structure itself remains difficult to interrogate. Here, we utilise focused ion beam – scanning electron microscopy nanotomography (FIB-SEM-NT) to quantitatively characterise the morphology of printed nanostructured networks and their devices using nanometre-resolution 3D images. The influence of nanosheet/nanowire size on network structure in printed films of graphene, $WS_2$ and silver nanosheets (AgNSs), as well as networks of silver nanowires (AgNWs), is investigated. We present a comprehensive toolkit to extract morphological characteristics including network porosity, tortuosity, specific surface area, pore dimensions and nanosheet orientation, which we link to network resistivity. By extending this technique to interrogate the structure and interfaces within printed vertical heterostacks, we demonstrate the potential of this technique for device characterisation and optimisation.

Liquid-deposited networks of 0D nanoparticles, 1D nanowires or nanotubes and 2D nanosheets have shown great promise across emerging applications in printed electronics[1,2], sensing[3], catalysis[4] and energy storage[5]. In particular, devices based on networks of 2D materials have been the subject of intensive research due to the electronic diversity of such materials[6], as well as recent advances in both scalable nanosheet production and deposition techniques[7]. An array of devices based on printed nanosheet networks have been demonstrated including transistors[8], capacitors[9], photodetectors[10], sensors[11] and supercapacitors[12]. However, it has become clear that the performance of these devices is almost always limited by network morphology[13].

Printed 2D networks tend to consist of porous, disordered arrays of nanosheets with variable degrees of connectivity,

alignment, and inter-sheet coupling. These morphological factors have been shown to heavily influence carrier mobility in nanosheet devices. Printed networks of poorly-aligned $MoS_2$ nanosheets[14] demonstrate values of $\approx 0.1$ cm$^2$ V$^{-1}$ s$^{-1}$, while spin-coated networks of conformally tiled nanosheets[15] exhibit mobilities of $\approx 10$ cm$^2$ V$^{-1}$ s$^{-1}$. In printed 2D capacitors and transistors the morphological tailoring of dielectric layers is crucial to ensure spatial continuity and prevent interlayer electrical shorting[16]. Alternatively, network porosity and pore tortuosity determine the degree of accessible nanosheet surface area for sensing or catalysis[17,18], as well as electrolyte infiltration and ion kinetics in battery and supercapacitor electrodes[19]. Despite the fundamental role morphology plays in maximising the physical properties of such solution-processed devices, optimising their

[1]School of Physics, CRANN and AMBER Research Centres, Trinity College Dublin, Dublin 2, Ireland. [2]Department of Physics, Faculty of Sciences, University of Oviedo, C/ Leopoldo Calvo Sotelo, 18, 33007 Oviedo, Asturias, Spain. [3]These authors contributed equally: Cian Gabbett, Luke Doolan. ✉e-mail: colemaj@tcd.ie

performance remains limited by the lack of morphological characterisation.

Standard techniques such as mercury intrusion porosimetry (MIP) and $N_2$ BET analysis have been used to determine the pore-size-distribution and specific surface area in thick, vacuum filtered nanosheet networks[20]. However, these methods generally require sample volumes (film thicknesses > 100 μm[20]) that are far beyond the scope of printed thin-film devices. Furthermore, such techniques can require high-temperature annealing pre-treatment steps[21], which are incompatible with temperature-sensitive flexible substrates[22], while sample preparation[23] and high-pressure mercury intrusion can irreversibly alter the network structure[24]. Atomic force microscopy (AFM) and SEM can only provide surface information, although SEM can also analyse cross-sections. Established 3D imaging techniques such as X-ray computed tomography (X-ray CT) or electron tomography (3D TEM) are routinely used to characterise samples for metrological[25], biological[26] and materials science applications[27,28]. However, these techniques can require non-trivial sample preparation[29,30] and there is a trade-off between the sampled volume and spatial resolution. X-ray nano CT is used to probe larger sample volumes than FIB-SEM-NT but is limited by voxel sizes on the order of tens of nanometres[31]. This has been shown to be inadequate for characterising the morphology of nanosheet networks, as pores or nanosheets less than ≈50 nm in size cannot be resolved[32]. Insufficient spatial resolution can also cause interphase boundaries to become blurred, which has been linked to underestimations in the measured pore connectivity and specific surface area in battery electrodes[33]. Alternatively, 3D TEM can offer sub-nanometre resolutions[34] for electron transparent samples ( < 500 nm thick)[26]. However, this is at the cost of drastically reduced sample volumes of ≈1 μm³ [35]. Volumes this size are not expected to be representative of a printed nanosheet network where the constituent nanosheets are often > 500 nm in length, or printed heterostacks with thicknesses >1 μm. FIB-SEM-NT effectively bridges the gap between these tomographic techniques by offering spatial resolutions of a few nanometres over representative sample volumes of $10^2 - 10^4$ μm³ [36]. This has been demonstrated through high-resolution reconstructions of oil shales[37], drug release coatings[38], fuel cells[39] and commercial battery electrodes[40].

Here, we utilise FIB-SEM-NT to interrogate the morphology of printed nanostructured networks at high resolution. We report 3D imaging with a voxel size of 5 nm × 5 nm × 15 nm and demonstrate a suite of techniques to extract quantitative morphological information from these images. We apply FIB-SEM-NT to characterize network structure in printed graphene, $WS_2$ and AgNS films, as well as AgNW networks, finding the morphological properties to scale with nanosheet or nanowire dimensions. This is then directly linked to the electrical resistivity of printed graphene networks of different nanosheet sizes. We extend this analysis to compare printed networks of graphene nanosheets produced using electrochemical exfoliation (EE) and liquid phase exfoliation (LPE), and to quantitatively analyse interfaces within printed vertical heterostacks. Finally, we demonstrate a machine-learning protocol to further enhance resolution in FIB-SEM-NT produced 3D volumes by generating intermediate network images and cubic voxels.

## Results

### 3D imaging using FIB-SEM nanotomography

LPE yields suspensions of nanosheets in various solvents (Fig. 1a) which can then be printed into networks[41,42]. Conventional surface and cross-sectional SEM imaging of a spray-cast graphene nanosheet network (nanosheet length, $l_{NS}$ = 695 nm, Fig. 1b) allows qualitative observations of network porosity or nanosheet alignment. However, it is difficult to extract quantitative information. Here, we use FIB-SEM-NT to produce high-resolution 3D reconstructions of portions of the network, which we refer to as 3D imaging. To achieve this (Fig. 1c), ≈800 network cross-sections are sequentially milled using the FIB and imaged using the SEM (Supplementary Note 2). Each network slice has an in-plane pixel size of 5 nm and average thickness of ≈15 nm, giving a voxel volume of ≈375 nm³. While this is a destructive technique that removes material from the network, it can confer voxel sizes 10–1000 times smaller than conventional X-ray CT scanners[43,44].

To enable quantitative analysis, each image in the stack is classified into its pore and nanosheet components using trainable WEKA segmentation[45] (Supplementary Note 3). All images were captured at an accelerating voltage of 2 kV using the SE2 detector to alleviate

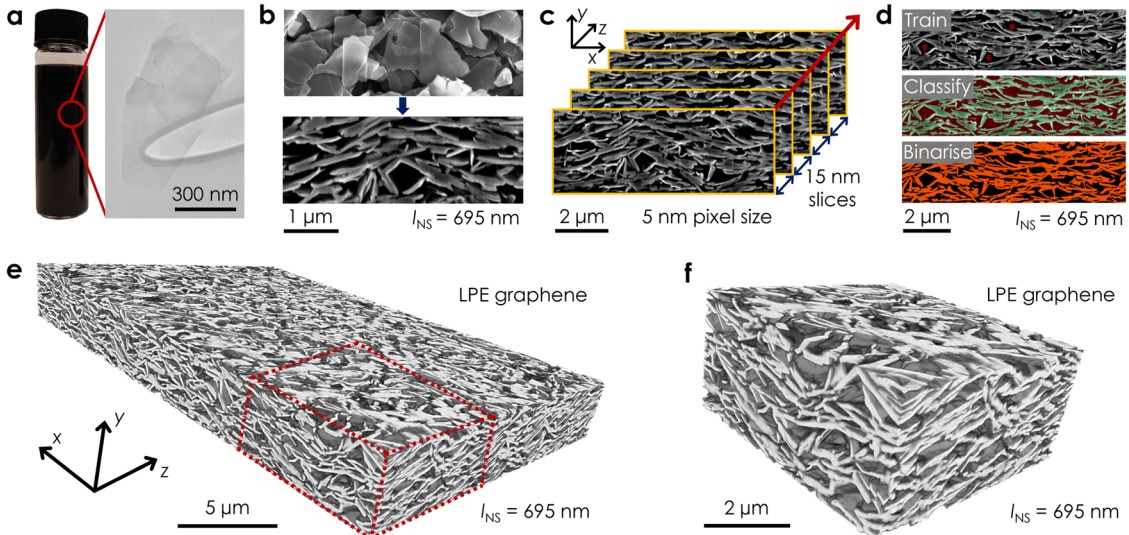

**Fig. 1 | FIB-SEM nanotomography of a printed LPE graphene network. a** Photo of a graphene dispersion and a typical TEM image of a liquid-exfoliated nanosheet. **b** Representative surface and cross-sectional SEM images of a printed multilayer graphene nanosheet network (nanosheet length, $l_{NS}$ = 695 nm). **c** Schematic of the slice-and-scan process. Network cross-sections are sequentially milled and imaged to produce a stack of 15 nm thick slices. The red arrow denotes the milling direction. **d** Image segmentation pipeline to classify a greyscale network cross-section into its nanosheet and pore components. Regions identified as nanosheets/pores by the user are indicated by green/red circles in Train, with the output of the classification process shown in Classify. The segmentation result is shown in Binarise, where the orange and black pixels correspond to regions labelled as nanosheets and pores respectively. **e** 3D reconstruction of a printed LPE graphene network generated using FIB-SEM-NT. **f** Magnified region of **e** showing nanosheets and pores.

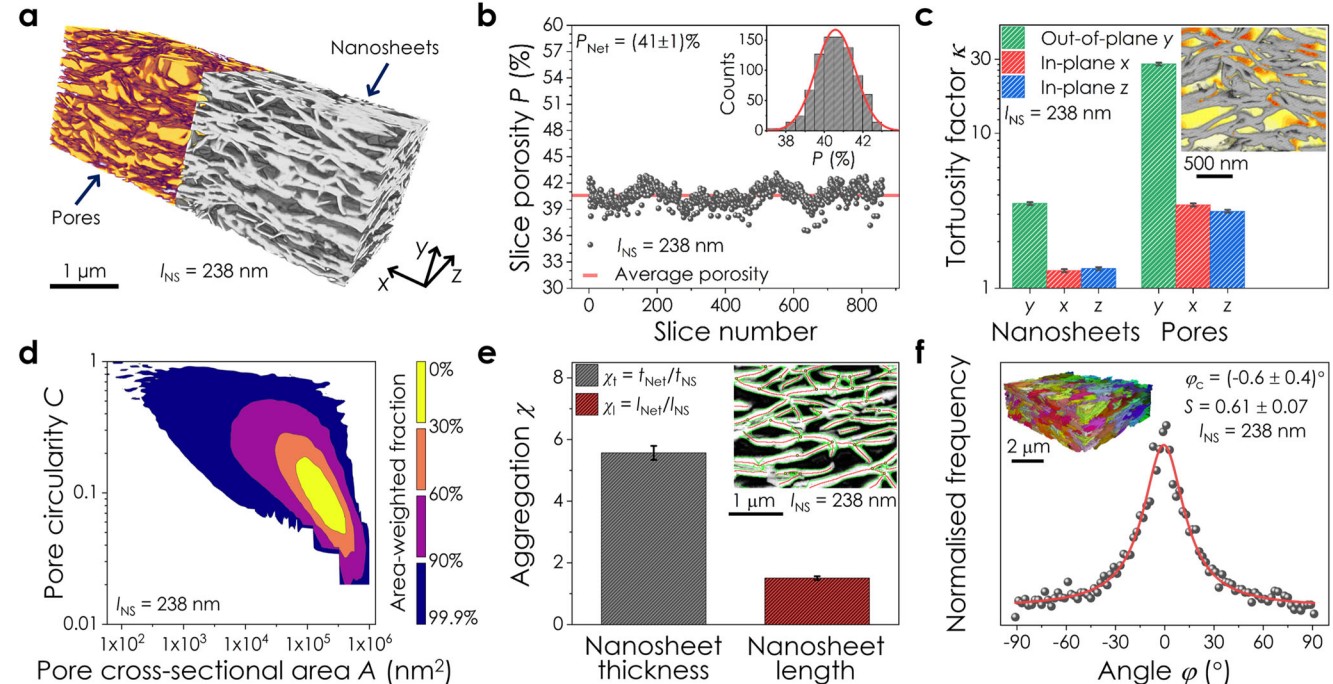

**Fig. 2 | Quantitative analysis of a reconstructed LPE graphene network.**
**a** Reconstructed portion of a printed graphene network ($l_{NS} = 238$ nm) separated into its pore and nanosheet contributions. **b** Porosity measurements for each slice in the network. The red line is the average porosity, $P$, across the 3D volume. Inset: histogram of slice porosity values. **c** Measurements of the tortuosity factor, $\kappa$, in the out-of-plane ($y$) and in-plane ($x,z$) directions for the nanosheet and pore volumes. Inset: normalised local flux through the network pore volume in the imaging ($z$) direction. Nanosheets are coloured grey and darker regions of the pore volume represent bottlenecks in the diffusive flux. $\kappa$ is calculated through a volume of 536 μm³ with an uncertainty of ± segmentation error. **d** Pore circularity, $C$, plotted as a function of cross-sectional area, $A$, for each pore chamber in the network. The contours denote the percentage of the total pore area contained within each band. **e** Nanosheet aggregation factors in both length, $\chi_l$, and thickness, $\chi_t$, during

network deposition. Nanosheet dimensions in the ink are given by ($l_{NS}$, $t_{NS}$), while ($l_{Net}$, $t_{Net}$) are the restacked nanosheet length and thickness in the network. Inset: nanosheet dimensions measured from network cross-sections. Pores and nanosheets are coloured black and white respectively. The estimated nanosheet length ($l_{Net}$) is denoted by red contours, inter-sheet junctions by hollow circles, and the nanosheet thickness ($t_{Net}$) by green lines. The data are presented as means ± the root sum of squares (RSS) of segmentation error and standard errors (SE) in the mean for ($l_{NS}$, $t_{NS}$) ($n = 190$) and ($l_{Net}$, $t_{Net}$) ($n > 10^5$). **f** Distribution of angles ($\varphi$) between the nanosheet normal vectors and the out-of-plane ($y$) direction in the network. The solid line is a fit to a Cauchy-Lorentz distribution centred on $\varphi_C = -0.6°$. Inset: discrete colour-coded nanosheets isolated using a 3D distance transform watershed. Source data are provided as a Source Data file.

sample charging and shine-through effects[46]. This ensures that only nanosheets and pores at the cross-section face are considered for classification (Supplementary Fig. 12). As shown in Fig. 1d, regions of each slice are first manually assigned as either nanosheet or pore, providing training data for the classifier. Each slice is then segmented into these components on a pixel-by-pixel basis using a random forest classifier[47] to create a binary image. These segmented image stacks are then aligned and interpolated in Dragonfly (Supplementary Note 4) to form a 3D network reconstruction (Fig. 1e). A typical network volume of 20 μm × 15 μm × 2 μm contains ≈1.6 × 10⁹ voxels. This allows the morphological properties of the network to be characterised on a voxel-by-voxel basis, as shown for a printed LPE graphene network ($l_{NS} = 695$ nm) in Fig. 1e, f. Crucially, FIB-SEM-NT facilitates analysis of these networks over representative volumes (Supplementary Note 5) and at a resolution that preserves the discrete nanosheet and pore components (Fig. 1f and Supplementary Movie 1).

**Analysing 3D images of nanosheet networks**
A printed graphene network ($l_{NS} = 238$ nm) that has been split into its pore and nanosheet volumes is shown in Fig. 2a. The network porosity, $P$, determines the nature of the intersheet junctions and influences rate performance in 2D battery electrodes[48]. However, due to sample and resolution limitations, this is often determined from sample weighing[14] or qualitatively discussed using SEM cross-sections. Here, the porosity of the printed graphene network was measured both across the entire volume and on a slice-by-slice basis using FIB-SEM-NT (Fig. 2b and

Supplementary Note 6)[49,50]. While the global porosity was calculated to be 41 ± 1%, the scatter in the slice-by-slice data highlights the local inhomogeneity of the network.

Pore connectivity is a key parameter that determines the accessible nanosheet surface area in sensing or electrochemical applications[51,52]. The pore volume in Fig. 2a was found to be highly contiguous, consistent with MIP/BET data for filtered graphene networks[20], with >99% of the total pore volume contained in a single macropore spanning the network. Measurements across different regions of the 3D volume revealed a connectivity length-scale of ≈251 nm. This represents the required depth in the imaging ($z$) direction for discrete pores to coalesce into a highly-connected pore volume (Supplementary Note 6)[50]. To quantitatively assess the pore and nanosheet connectivity we calculated the tortuosity factor[53], $\kappa$, which is related to tortuosity, $\tau$, through $\kappa = \tau^2$. This was determined by measuring the reduction of diffusive flux through each network in the in-plane (IP, $x$, $z$) and out-of-plane (OOP, $y$) directions using TauFactor[49] (Fig. 2c). A tortuosity factor of $\kappa = 1$ represents an unobstructed path through the pore/nanosheet volume in a given direction, while values >1 arise from a convolution of the network structure. The nanosheet network tortuosity factor influences charge transport through the film. Charge carriers in more well-packed networks experience less tortuous paths with fewer, more conformal junctions[2]. Pore tortuosity heavily affects rate performance in nanosheet-based battery electrodes[54], while in gas sensing applications the porosity, pore size and tortuosity are directly linked to gas

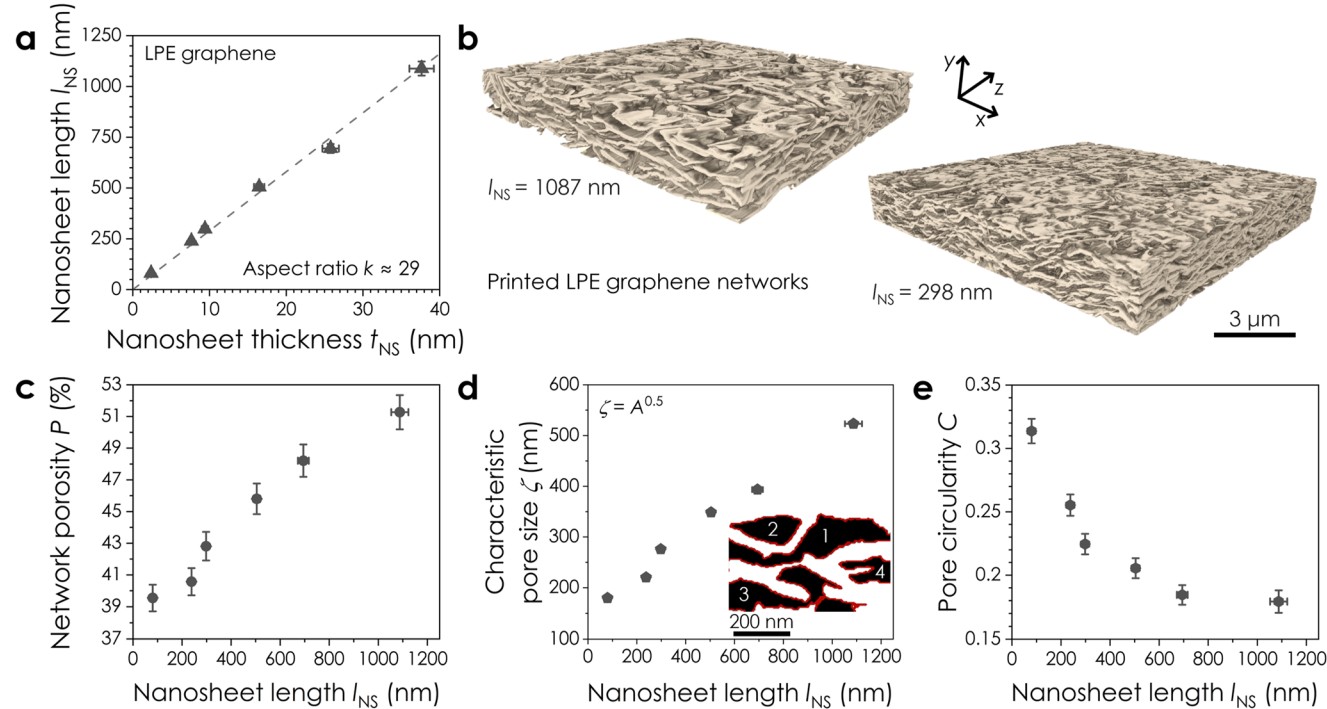

**Fig. 3 | Porosity in printed graphene networks as a function of nanosheet size.**
**a** Relationship between the nanosheet length, $l_{NS}$, and thickness, $t_{NS}$, for each size-selected 2D ink. The data are presented as means ± SE in the mean ($n = 190–270$).
**b** Reconstructed 3D volumes for printed networks comprised of large ($l_{NS} = 1087$ nm) and small ($l_{NS} = 298$ nm) LPE graphene nanosheets. **c** Plot of network porosity, $P$, as a function of $l_{NS}$. **d** Scaling of the mean pore size, $\zeta = \sqrt{A}$, in each network with $l_{NS}$, where $A$ is the average pore cross-sectional area. Inset: pore

identification and labelling in FIJI, where black and white pixels correspond to pores and nanosheets respectively. Each discrete pore is numbered and outlined by a solid red line. **e** Plot of the pore circularity, $C$, for each network as a function of $l_{NS}$. The data are presented as means ± the RSS of segmentation error and SE in the mean for $P$ ($n = 605–860$), $\zeta$ ($n = 145 – 438$) and $C$ ($n = 145–438$). Source data are provided as a Source Data file.

diffusion[55]. Bottlenecks in the diffusive flux through the pore volume are shown visually in Fig. 2c. The comparatively smaller nanosheet $\kappa$-values in Fig. 2c are reflective of a well-connected nanosheet network, while $\kappa_{OOP}/\kappa_{IP} > 1$ suggests that the nanosheets are primarily aligned in-plane. This is consistent with directional anisotropies in conductivity and mass transport through solution-processed 2D networks[56].

Because the pore volume is highly connected, we analyse the cross-sectional area and shape of 2D pore sections in each slice (Supplementary Note 8). The pore circularity, $C$, is plotted as a function of pore cross-sectional area, $A$, for each pore in the network in Fig. 2d, where $C = 4\pi A$/perimeter$^2$. Here, a value of $C = 1$ represents a circular pore cross-section while smaller values correspond to more irregular and elongated pores. The area-weighted heat map suggests that the pore volume is dominated by pore chambers with cross-sectional areas $>10^4$ nm$^2$, and that larger pores have elongated geometries, consistent with published BET measurements[20].

It is well-reported that solution-processed nanosheets tend to restack during deposition[57]. We determined the degree and nature of this restacking by measuring the nanosheet length and thickness in the ink ($l_{NS}$, $t_{NS}$) using AFM, as well as the aggregated nanosheet dimensions in the network ($l_{Net}$, $t_{Net}$) post-deposition. The restacked nanosheet length and thickness were measured from network cross-sections using the Ridge Detection plugin in FIJI[50,58] (Fig. 2e, inset, and Supplementary Note 9). We define the aggregation factors in nanosheet length, $\chi_l$, and thickness, $\chi_t$, as $\chi_l = l_{Net}/l_{NS}$ and $\chi_t = t_{Net}/t_{NS}$ respectively. Values of $\chi_l \approx 1.5$ and $\chi_t \approx 5.6$ were found for the printed LPE graphene network in Fig. 2e. This is in agreement with a value of $\chi_t \approx 5$ reported for vacuum filtered WS$_2$ networks[59], and suggests that nanosheets primarily aggregate through vertical restacking with maximised basal plane overlap.

By isolating discrete nanoplatelets and noting their orientation (Fig. 2f, inset, and Supplementary Note 10)[60], the distribution of angles, $\varphi$, between each nanoplatelet's normal vector and the out-of-plane ($y$) direction was calculated. The data in Fig. 2f was fit with a Cauchy-Lorentz distribution centred on $\varphi_C \approx -0.6°$, which suggests the nanosheets are primarily aligned in the plane of the film. The full width at half maximum (FWHM) of the distribution provides an estimate of the degree of alignment about $\varphi_c$ in the network[61]. The FWHM of $(29 \pm 1)°$ for the spray cast network in Fig. 2f is comparable to a value of 21° for an inkjet-printed graphene film measured using AFM. In addition, we measured the Hermans orientation factor[62], $S = (3\langle \cos^2 \varphi \rangle - 1)/2$, to be $0.61 \pm 0.07$ for the network, which is consistent with partial in-plane alignment. A value of $S = 1$ would imply the nanosheets are perfectly aligned in the plane of the film, while $S = 0$ for randomly oriented nanosheets. This is in broad agreement with a value of $S = 0.79$ for a vacuum filtered Ti$_3$C$_2$T$_x$ nanosheet network measured using wide-angle X-ray scattering (WAXS)[32].

## Characterising size-dependent morphology

The physical properties of 2D networks are known to scale with nanosheet size[63,64]. Here, we use FIB-SEM-NT to systematically study the morphology of printed LPE graphene networks for various nanosheet lengths, $l_{NS}$. Size-selected inks were produced using liquid cascade centrifugation[65], characterised by AFM (Fig. 3a) and spray-coated into networks. Reconstructed 3D volumes for networks of two different nanosheet sizes in Fig. 3b show noticeable changes in network morphology as $l_{NS}$ is decreased from 1087 to 298 nm. Analysis reveals a clear decrease in network porosity from 51% to 39% with decreasing $l_{NS}$ (Fig. 3c), with a corresponding reduction in the characteristic pore size, $\zeta = \sqrt{A}$, in Fig. 3d. The pore circularity data similarly exhibits a dependence on $l_{NS}$ (Fig. 3e), where networks of smaller

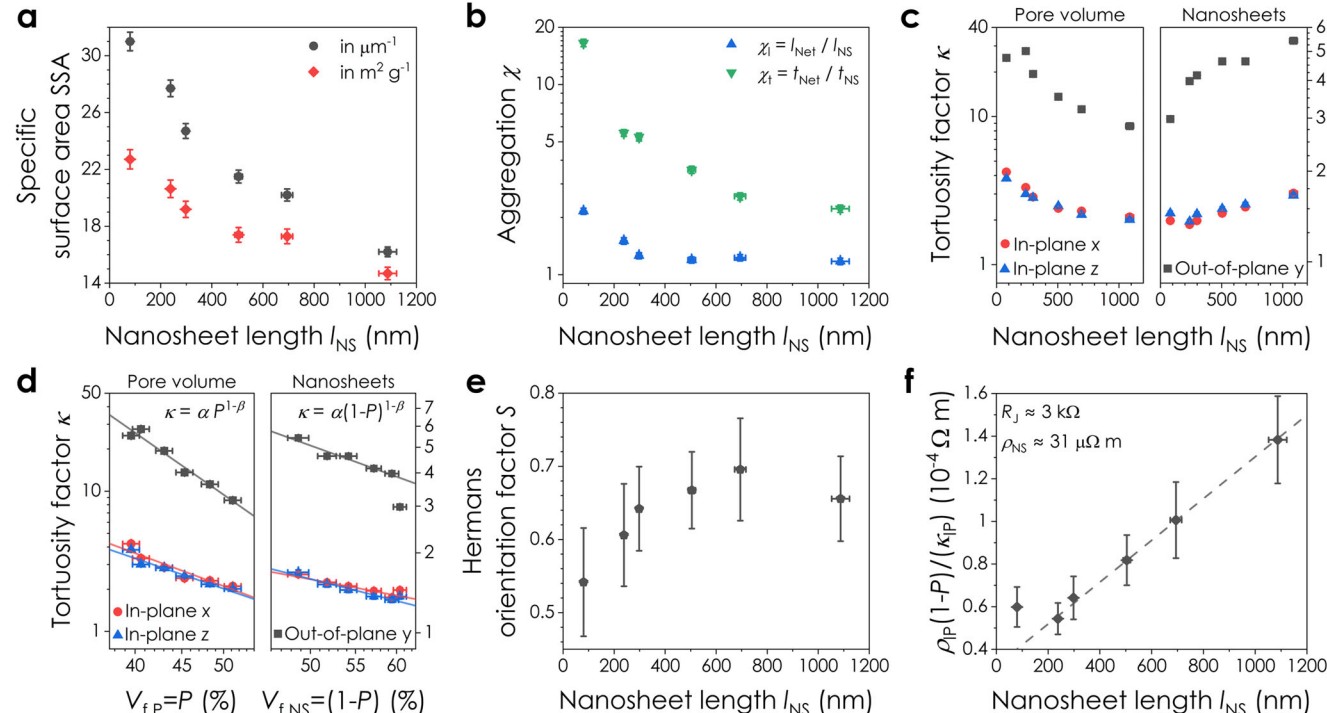

**Fig. 4 | Morphology of printed graphene networks as a function of nanosheet size. a** Network specific surface area plotted against $l_{NS}$ in ($\mu m^2\ \mu m^{-3}$) and ($m^2\ g^{-1}$) units. SSA is calculated from network volumes of 279 – 706 $\mu m^3$ with an uncertainty of ± segmentation error. **b** Nanosheet aggregation factors in thickness, $\chi_t$, and length, $\chi_l$, post-deposition plotted as a function of nanosheet length. The data are presented as means ± the RSS of segmentation error and SE in the mean for ($l_{NS}$, $t_{NS}$) ($n = 190 - 270$) and ($l_{Net}$, $t_{Net}$) ($n > 9000$). **c** Plot of the pore and nanosheet tortuosity factors in the out-of-plane ($y$) and in-plane ($x,z$) directions against $l_{NS}$. **d** Plot of the pore and nanosheet tortuosity factors in each direction as a function of the volume fraction of pores ($P$) and nanosheets (1-$P$). The solid lines are fits to an

adjusted Bruggeman relation described by $\kappa = \alpha P^{1-\beta}$ for the pore data and $\kappa = \alpha(1 - P)^{1-\beta}$ for the nanosheets, where $\alpha$ is a prefactor and $\beta$ is the fitted Bruggeman exponent. The uncertainty in $\kappa$ is ± segmentation error. **e** Hermans orientation factor, $S$, plotted as a function of nanosheet length for each network. The data are presented as means ± the upper and lower bounds of the 3D distance transform watershed ($n > 1800$, Supplementary Note 10). **f** Plot of the morphologically scaled network resistivity ($\rho_{IP}(1 - P))/\kappa_{IP}$ against $l_{NS}$, where $\rho_{IP}$ is the in-plane electrical resistivity and $\kappa_{IP}$ is the in-plane tortuosity factor of the nanosheets. The straight line is a fit to Eq. 1. The data are presented as means ± SE in the mean ($n = 9$). Source data are provided as a Source Data file.

nanosheets have more circular and compact pore cross-sections. This implies that printed networks comprised of smaller nanosheets are more densely packed, which has been linked to improved charge transfer in graphene films[66]. Alternatively, networks of larger nanosheets are more open and porous, facilitating enhanced electrolyte infiltration and mass transport. Taken together, the data in Fig. 3c-e suggests that changing the nanosheet size offers a simple means to tailor the network porosity for a target application. FIB-SEM-NT can be used to inform this by measuring pore sizes that span from a few nanometres to microns.

The dependence of network morphology on $l_{NS}$ is further reflected in the specific surface area (SSA) of the networks (Fig. 4a). This is a key parameter that describes the accessible nanosheet surface area for sensing, catalytic and energy storage applications[67]. Here, the network SSA is seen to decrease from 23 to 15 $m^2\ g^{-1}$ as $l_{NS}$ increases from 80 to 1087 nm. This is consistent with expectations that SSA scales inversely with nanosheet thickness[68], $t_{NS}$, given $t_{NS}$ and $l_{NS}$ are intrinsically coupled for LPE[69], and agrees with a value of 25 $m^2\ g^{-1}$ for $V_2O_5$ nanosheet films measured using BET[70]. Such low values of SSA (cf. 2600 $m^2\ g^{-1}$ for graphene monolayers[71]) are often attributed to restacking during deposition[72]. By comparing the dimensions of the aggregated nanosheets in each network ($l_{Net}$, $t_{Net}$) to the dimensions measured by AFM in the inks ($l_{NS}$, $t_{NS}$), we quantified the degree of aggregation in length, $\chi_l$, and thickness, $\chi_t$, as a function of $l_{NS}$ (Fig. 4b). In each network $\chi_t$ was considerably larger than $\chi_l$. This suggests that the dominant aggregation mechanism during processing is vertical restacking of the nanosheets, with an increase in aggregation observed for smaller nanosheets in Fig. 4b.

The tortuosity factor, $\kappa$, of both the pore and nanosheet volumes, is plotted as a function of $l_{NS}$ in Fig. 4c. Both the nanosheet and pore tortuosity factors are significantly larger in the out-of-plane ($y$) direction due to nanosheet in-plane alignment. Interestingly, while the pore volume was found to become less tortuous with increasing $l_{NS}$, the nanosheet network exhibited the opposite trend, implying improved network connectivity for smaller LPE nanosheets. This is because the tortuosity factor of a given network is known to scale with the fractional volume occupied by that network[73]. This leads to a well-defined relationship between tortuosity factor and network volume fraction ($V_f$) for both the pore ($V_{f,P} = P$) and nanosheet ($V_{f,NS} = 1-P$) networks in Fig. 4d. The data follow an adjusted Bruggeman relation[74], $\kappa = \alpha V_f^{1-\beta}$, where $\alpha$ is a prefactor and $\beta$ is the Bruggeman exponent. The extracted exponents for pores ($\beta_{IP} = 3.3$ and $\beta_{OOP} = 5.6$) and nanosheets ($\beta_{IP} = 1.9$ and $\beta_{OOP} = 2.5$) are considerably larger than values of 1.5 predicted by basic models[73]. However, experimentally measured exponents[75] are usually >1.5, with values of $\beta = 2$-5 predicted for high-aspect ratio particles[76].

To characterise the nanosheet orientation with changing nanosheet size, we calculated the Hermans orientation factor, $S$, of each network. The values of $S = 0.54 - 0.7$ shown in Fig. 4e again imply in-plane nanosheet alignment. This is in agreement with values of $S = 0.67 - 0.87$ reported for vacuum-filtered graphene oxide (GO) networks measured using small-angle X-ray scattering (SAXS)[77]. Furthermore, the orientation factor appears to increase with increasing nanosheet length, $l_{NS}$ (Fig. 4e). This contrasts with solution-processed GO networks, where increased aspect ratios are known to drive improved alignment[61,63]. However, LPE nanosheets are comparatively

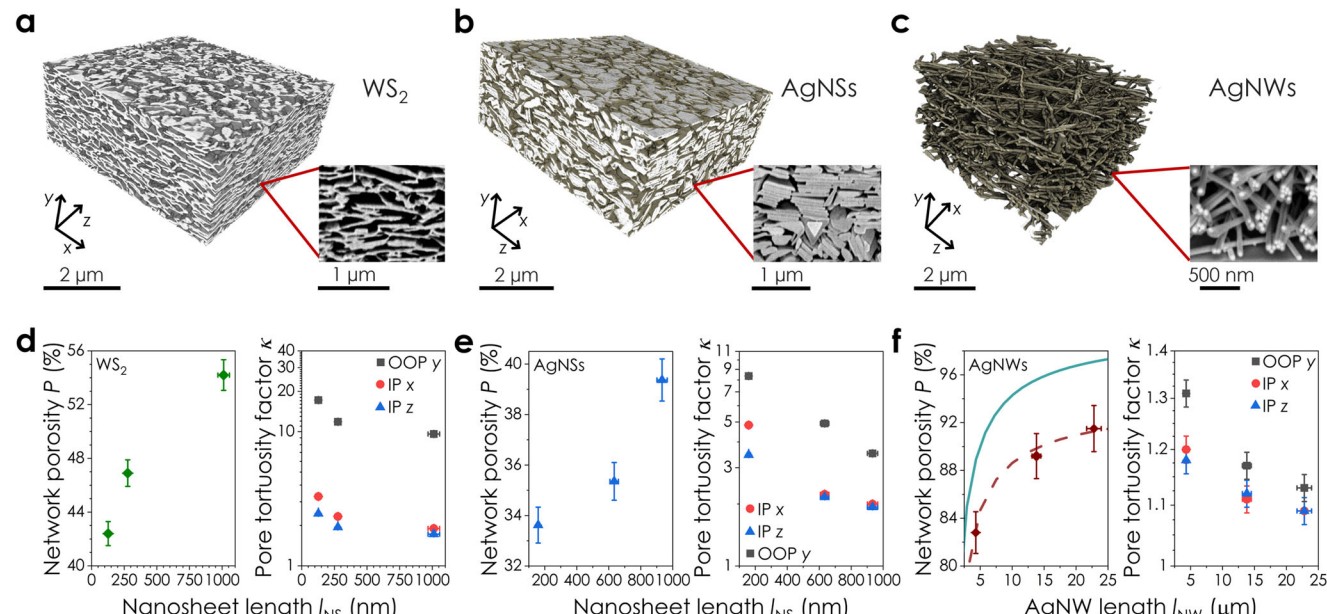

**Fig. 5 | Morphological characterisation of materials beyond graphene.** Reconstructed network volumes of printed **a** WS$_2$ nanosheets, **b** Silver nanosheets (AgNSs), and **c** Silver nanowires (AgNWs). Inset: representative FIB-SEM cross-sections for each material. **d**–**f** Plots of the network porosity and pore tortuosity factor, $\kappa$, as a function of nanosheet/nanowire length for the **d** WS$_2$, **e** AgNS and **f** AgNW networks. The solid line in **f** is the predicted porosity scaling with AgNW length[79], which we find to describe our data well when scaled by a prefactor of 0.94 (dashed line). The uncertainty in $l_{NS}$ ($n = 139 – 375$) and $l_{NW}$ ($n = 40-50$) is ± SE in the mean. The error bars in $P$ represent the RSS of segmentation error and SE in the mean ($n = 175 – 701$). $\kappa$ is calculated from network volumes of 212–921 $\mu m^3$ with an uncertainty of ± segmentation error. Source data are provided as a Source Data file.

smaller with roughly constant aspect ratios driven by nanosheet mechanics[69]. Thus, we propose that smaller, thinner nanosheets conform to each other more easily[78], leading to a reduction in network porosity while increasing the angular dispersion.

The network morphology should strongly impact electrical properties in printed nanosheet networks. As shown in Supplementary Note 11, the in-plane resistivity of a nanosheet network, $\rho_{IP}$, is dependent on $P$, $l_{NS}$ and $\kappa_{IP}$, as well as the junction resistance, $R_J$, nanosheet resistivity, $\rho_{NS}$, and aspect ratio, $k_{NS}$:

$$\rho_{IP} \approx \frac{\rho_{NS} + (R_J l_{NS}/k_{NS})}{(1-P)/\kappa_{IP}} \qquad (1)$$

We measured the network electrical resistivity as a function of $l_{NS}$, with the data plotted in a linearised form in Fig. 4f. A straight line fit yields reasonable values[13] of $R_J \approx 3$ k$\Omega$ and $\rho_{NS} \approx 31$ $\mu\Omega$ m, highlighting the dependence of network resistivity on morphological factors. This demonstrates that the structure and effective mobility of a network can be tuned by changing the size of its constituent nanosheets. Furthermore, we show in Supplementary Fig. 37 that the specific contact resistivity of the printed networks decreases as $l_{NS}$ is reduced and the nanosheet volume fraction increases. Crucially, these morphological changes can be quantitatively measured using FIB-SEM-NT and used to inform device optimisation.

## Versatility of 3D imaging

FIB-SEM nanotomography is a general method to analyse nanoscale networks. To show this, we 3D imaged printed networks of WS$_2$ nanosheets, AgNSs and AgNWs (Fig. 5a–c). While there are distinct morphological differences between these systems, there are also similarities. Increasing the nanosheet/nanowire length causes the network porosity and pore tortuosity factor to increase and decrease respectively (Fig. 5d–f), similar to the printed graphene networks. However, the magnitude of this porosity change is somewhat smaller for AgNSs than in WS$_2$ (or graphene) networks. These differences in

porosity scaling with $l_{NS}$ (Supplementary Fig. 20) suggest that there are material-dependent factors contributing to the network morphology. Indeed, while the ratio of ($\kappa_{OOP}/\kappa_{IP}$) for nanosheets in the AgNS and WS$_2$ networks was found to be ≈1.5 and ≈2.5 respectively, inferring considerable in-plane alignment, they are each different from the value of ≈2.9 found for graphene networks (Supplementary Fig. 21). Furthermore, the aligned restacking previously discussed for graphene can clearly be seen in the AgNS networks, where vertical stacks of 3 − 7 AgNSs are visible (Fig. 5b, inset). A similar effect can be inferred for the WS$_2$ nanosheets (Supplementary Fig. 32).

The 3D image of the AgNW network (Fig. 5c) allows AgNWs to be resolved both as isolated 1D objects and within small bundles. This highlights the resolution advantage of FIB-SEM-NT, as the AgNW diameter of ≈55 nm is smaller than the pixel size for many X-ray CT techniques[31]. As with the nanosheet data, the porosity of an AgNW network is seen to increase with increasing AgNW length. This is consistent with a modified version of a model reported for randomly packed rigid fibres[79] (Fig. 5f). The nanowire ($\kappa_{OOP}/\kappa_{IP}$) ratio of ≈12 in the AgNW networks (Supplementary Fig. 21) is much higher than for the 2D nanosheet volumes. This is driven by the considerably higher porosities of $P = 82\%$ – 91% and strong in-plane alignment for the AgNWs, as shown in Fig. 5c. Taken together, the data in Fig. 5a–f highlights the applicability of FIB-SEM-NT to characterise networks of 1D and 2D nanomaterials with feature sizes that range from a few nanometres to tens of micrometres. This is summarised for the materials studied in this work in Table 1.

## Characterising electrical and device properties

It has been reported that networks of electrochemically exfoliated nanosheets[15] display much higher mobility than their LPE counterparts[14] for reasons of morphology[13]. Spin-coated networks of conformally tiled and high aspect ratio EE nanosheets have demonstrated basal plane separations <1 nm, which is only resolvable using cross-sectional TEM[15]. However, it is still possible to characterize the nanostructure and mesoporosity (pore sizes >5 nm) of such

highly-aligned networks using the spatial resolutions afforded by FIB-SEM-NT[2]. Here, we investigate the surface roughness, tortuosity factor, mesoporosity, and electrical conductivity of EE ($t_{NS}$ = 3.7 nm) and LPE ($t_{NS}$ = 20 nm) graphene nanosheet networks deposited under identical conditions. 3D imaging reveals significant differences in network structure (Fig. 6a). The porosity of the EE network ($P$ = 28%) is considerably lower than the value of $P$ = 44% for its LPE counterpart. This suggests that the EE nanosheets are more densely packed, which is reflected by in-plane tortuosity factors of $\kappa_{IP\text{-}NS}$ = 1.10 and $\kappa_{IP\text{-}NS}$ = 1.44 for the EE and LPE networks. Electrical measurements show the EE network to be ≈6 times more conductive (Fig. 6a). Combining the values in Fig. 6a with Eq. 1 (using $t_{NS} = l_{NS}/\kappa_{NS}$) and approximating

these networks as purely junction limited ($R_J \gg R_{NS} = \rho_{NS}/t_{NS}$), implies $R_J$ in these EE and LPE networks to be very similar. This means that against expectations, the conductivity disparity, in this case, is predominantly due to the nanosheet thickness difference rather than morphological factors ($P$, $\kappa_{IP}$, and $R_J$). By reconstructing the surface topography of both networks (Fig. 6a and Supplementary Note 12) we measured the root mean square roughness, $R_{RMS}$, of the EE and LPE networks to be ≈122 nm and ≈182 nm respectively. The reduced surface roughness in printed EE networks can improve interface quality in vertically stacked devices[80].

The nature of the interfaces within a printed vertical heterostack can significantly influence device reproducibility and performance[81]. To highlight this, we deposited LPE graphene networks of two different nanosheet lengths ($l_{NS}$ = 630 nm and 215 nm), which were then coated with silver nanoparticles (AgNP, diameter ≈50 nm) to mimic top electrode deposition. 3D imaging (Fig. 6b) could resolve individual nanoparticles, and by removing the nanosheet layer the incorporation of AgNPs into each network was assessed. Isolated silver nanoparticles were found to have penetrated ≈1.3 μm into the network of larger nanosheets ($l_{NS}$ = 630 nm). Connectivity analysis revealed that a percolating AgNP path reached ≈725 nm into the layer. However, virtually no AgNP penetration was found in the network of smaller nanosheets ($l_{NS}$ = 215 nm). This aligns with the data in Figs. 3b–d and 4c, which shows networks of smaller nanosheets to be more densely packed with more tortuous pore volumes. To interrogate the AgNP/graphene interface, we measured their respective volume fractions as a function of depth in the out-of-plane ($y$) direction from the top surface of each heterostack (Fig. 6b and Supplementary Note 12). The interface between the nanosheet and nanoparticle layers is sharper and more well-defined for the heterostack comprised of smaller nanosheets

### Table 1 | Overview of the nanomaterials characterised using FIB-SEM-NT

|    | Nanomaterial | Length | Thickness | Reconstructed network |
|----|----|----|----|----|
| 0D | Silver nanoparticles (AgNPs) | 50 nm | 50 nm | Fig. 6b |
| 1D | Silver nanowires (AgNWs) | 4.3–22.9 μm | 55 nm | Fig. 5c, f |
| 2D | LPE graphene | 80–1087 nm | 2.4 – 38 nm | Figs. 1–4 |
|    | EE graphene | 2.3 μm | 3.7 nm | Fig. 6a |
|    | WS$_2$ | 127–1013 nm | 17 - 45 nm | Fig. 5a, d |
|    | Silver nanosheets (AgNSs) | 159–934 nm | 48 – 100 nm | Fig. 5b, e |

Dimensions of the nanomaterials that were printed into networks and reconstructed using FIB-SEM-NT in this work. The dimensionality, length, and thickness of each material is given, as well as the location of its reconstructed network volume.

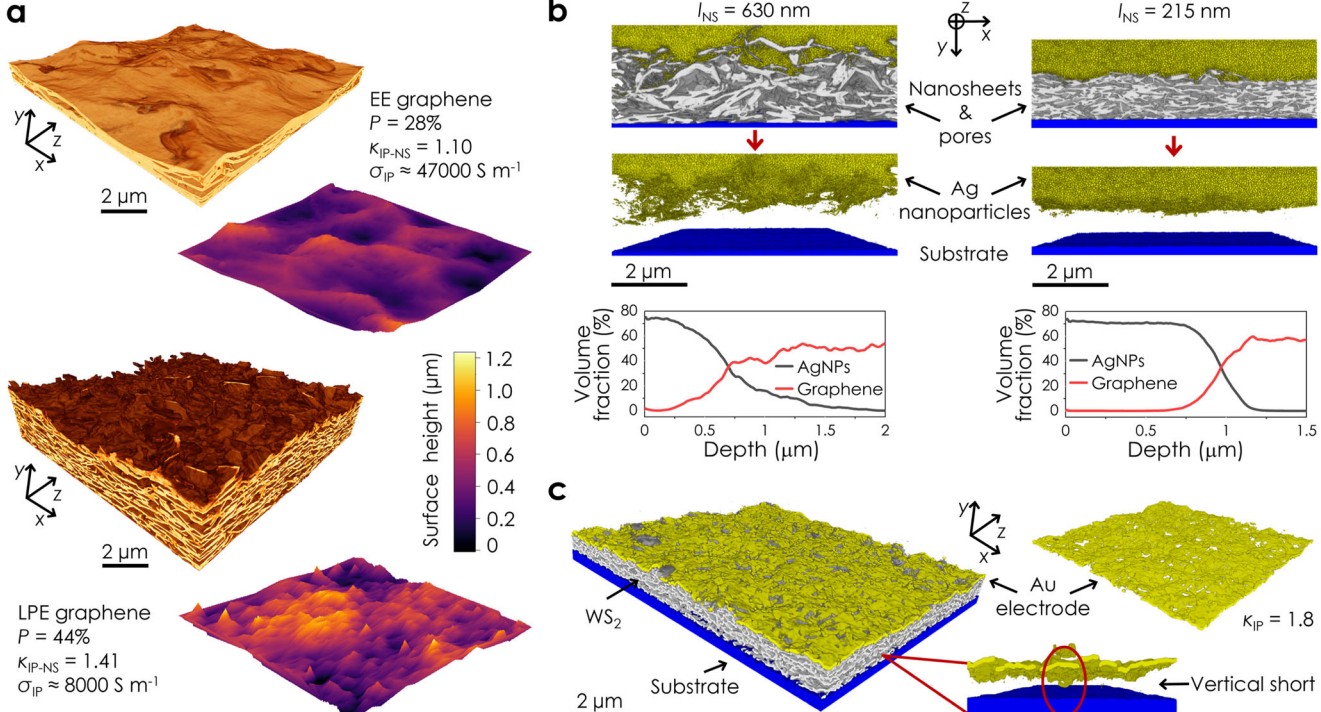

**Fig. 6 | Characterisation of electrochemically exfoliated graphene networks, vertical heterostacks, and nanostructured devices. a** 3D reconstructions and surface topography of printed EE and LPE graphene networks to compare the morphologies. The in-plane tortuosity factor, $\kappa_{IP\text{-}NS}$, conductivity, $\sigma_{IP}$, and porosity, $P$, of each nanosheet network is shown. **b** Printed graphene/silver nanoparticle (AgNP) stacks for two different nanosheet sizes ($l_{NS}$ = 215 and 630 nm) to characterise the degree of interlayer penetration with changing $l_{NS}$. The volume fraction

of each phase is plotted as a function of depth in the out-of-plane ($y$) direction from the top surface of each heterostack. **c** A reconstructed glass/ITO – WS$_2$ – evaporated Au device segmented into its discrete layers. Removal of the WS$_2$ nanosheet mid-layer allows vertical electrical shorts through the network to be identified. The Au top electrode has been isolated to show the presence of holes in the layer. Source data are provided as a Source Data file.

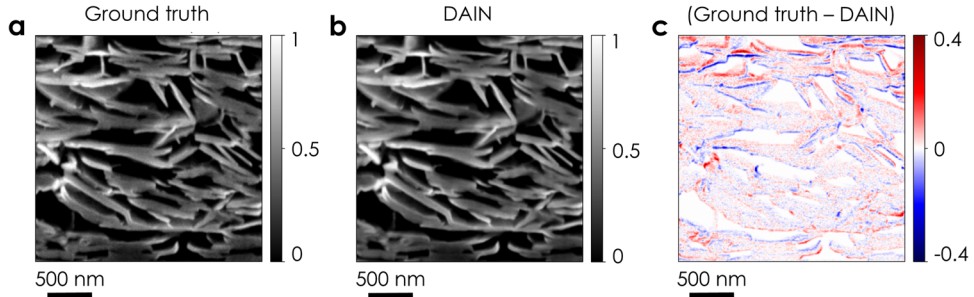

**Fig. 7 | Computer-generated intermediate images. a** A ground-truth image removed from the original stack of FIB-SEM data for a printed LPE graphene network. **b** The correspondent image generated by the video frame interpolation algorithm DAIN[85]. The colour scales in **a, b** represent the normalised pixel intensity in each image. **c** The intensity difference between the ground truth and DAIN generated images, where identical pixels have a value of zero (white). A positive (red) value indicates that pixels in the ground truth frame were brighter, while a negative (blue) value means that the ground truth pixel was darker.

$(l_{NS} = 215\,nm)$. This suggests that printed networks of smaller LPE nanosheets exhibit enhanced continuity and improved interface quality owing to their reduced porosity, smaller pores, more tortuous pore volumes, and decreased surface roughness. This can help mitigate charge trapping and pinhole formation, leading to improved performance in printed transistors and capacitors[82,83].

Moreover, this technique can be used to reconstruct complex devices, differentiating between various components. Shown in Fig. 6c is a 3D image of an ITO-Glass/WS$_2$/evaporated-Au vertical heterostack. Four-way segmentation allows this device to be separated into its discrete layers to enable analysis of the internal nanostructure. By removing the WS$_2$ nanosheets and pore volume, we can visualise the relationship between the substrate and gold, identifying electrical shorts between the top and bottom electrodes. Also, discrete device layers like the Au top electrode can be isolated and analysed individually (Fig. 6c). Here, the roughness of the underlying WS$_2$ network has caused holes to form leading to a poor-quality gold film. This is reflected by an in-plane tortuosity factor of $\kappa_{IP} = 1.8$, implying electrode resistances roughly double what might be expected.

### Generation of isotropic voxels

In this work, experimental constraints confine us to non-cubic (5 nm × 5 nm × 15 nm) voxels, which limits resolution and hinders analysis. Although one can produce cubic voxels by linear interpolation between adjacent frames[84], this yields image elements with blurred edges. Faced with similar problems, the computer-vision community utilises neural networks, such as DAIN[85]. These algorithms, usually trained on the Vimeo90K dataset[86], generate additional frames between consecutive images, increasing the resolution along the time direction. This strategy can improve the resolution along the milling direction of FIB-SEM-NT data by introducing intermediate slices between the imaged cross-sections. To test this approach, every second network cross-section was removed from a printed LPE graphene image stack, effectively doubling the slice thickness from 15 to 30 nm. The removed frames were then replaced with images generated by DAIN to allow for a direct comparison. This doubled the resolution in the milling direction and restored the original slice thickness of 15 nm using generated images. These can then be compared to the removed ground-truth frames, as displayed in Fig. 7a-c. We find good agreement, showing that neural-network-based approaches can be used to further enhance resolution in FIB-SEM-NT generated 3D images.

### Discussion

In summary, 3D imaging using FIB-SEM-NT allows the morphology of printed nanostructured networks to be quantitatively characterised with nanometre-resolution. This approach is versatile and can be applied to both conducting and semiconducting networks of 1D and 2D nanomaterials. We demonstrate the extraction of important morphological parameters from these 3D images and have applied this to systematically study the influence of nanosheet/nanowire dimensions on network structure. In addition, multi-phase segmentation allows FIB-SEM-NT to be extended to heterostacks and devices, where discrete layers and their interfaces have been analysed. We believe this technique will be an important tool to investigate and optimise a range of nano-enabled devices for emerging applications.

## Methods

### Ink preparation

Nanosheet inks were prepared using LPE[41]. Graphite powder (Asbury, Grade 3763) was first sonicated in 80 mL of deionised (DI) water (18.2 MΩ cm) at a concentration of 35 mg mL$^{-1}$ for 1 h. A horn probe sonic tip (Sonics Vibra-cell VCX-750 ultrasonic processor) at 55% amplitude, with a pulse rate of 6 s on and 2 s off was used. The resulting dispersion was centrifuged for 1 h at 2684× g (Hettich Mikro 220 R) to remove potential contaminants from the starting powder. The supernatant was decanted, and the sediment redispersed in 80 mL of DI water and sodium cholate (SC, Sigma Aldrich, >99%) at a concentration of 2 mg mL$^{-1}$. This was sonicated for 8 h at an amplitude of 55% with a 4 s on 4 s off pulse rate. The resulting dispersion was separated into inks of different nanosheet sizes using liquid cascade centrifugation[65]. An initial centrifugation step at 28× g for 2 h was used to isolate unexfoliated material in the sediment. The supernatant was then subjected to additional centrifugation steps at 112× g, 252× g, 447× g, 699× g, and 1789× g, retaining the sediment at each interval to isolate nanosheet fractions of different sizes. In each case the sediment was redispersed in a 2 mg mL$^{-1}$ DI:SC solution. The redispersed 112× g ink was subjected to a further centrifugation step at 28× g for 1 h to separate it into two size fractions. Each ink was then transferred to isopropanol (IPA, Sigma Aldrich, HPLC grade, 99.9%) for spray coating. To remove the sodium cholate, each dispersion was centrifuged for 2 h at 4052× g. The supernatant was discarded, and the sediment redispersed in IPA. This step was repeated twice. WS$_2$ inks were produced in a similar manner, however, the sonication of bulk powder (Alfa Aesar, 10 − 20 μm, 99.8%) was carried out in IPA. The size-selected inks were produced by centrifugation steps at 112× g, 252× g, 447× g, and 4052× g.

To facilitate the LPE vs. EE comparison, graphite powder was solvent exfoliated in N-methyl-2-pyrrolidone (NMP, Sigma Aldrich, HPLC grade, ≥ 99%) with a cleaning step as described above. The sediment was then redispersed in fresh NMP and exfoliated for 9 h at an amplitude of 55% with a 4 s on and 4 s off pulse rate. The produced dispersion was centrifuged at 447× g for 2 h to remove unexfoliated material and the supernatant was centrifuged at 1789× g for a further 2 h. The resulting sediment was transferred to IPA as described above to produce the LPE ink. To prepare the ink of electrochemically exfoliated graphene nanosheets, two pieces of graphite foil (Alfa Aesar, 254 μm thick, 99.8% metals basis) with dimensions 50 × 30 × 0.25 mm$^3$

were connected as anode and cathode to a DC power supply and immersed in 100 mL of an aqueous electrolyte solution of 0.1 M $(NH_4)_2SO_4$ (Alfa Aesar, > 98%) with a separation of 2 cm. A potential of 10 V was applied to the electrodes for 30 min, with a current that increased from ≈1 to ≈1.8 A during the process. The resulting expanded material in the electrolyte solution was filtered and repeatedly washed with DI water (≈1 L). This was then bath sonicated (Thermo Fisher Scientific, FB11201, 37 kHz) in 100 mL of dimethylformamide (DMF, Sigma Aldrich, HPLC grade, ≥ 99.9%) for 10 min to complete the exfoliation into a dispersion of graphene nanosheets. The resulting ink was centrifuged at 1006× $g$ for 20 min to remove incompletely exfoliated particles. The EE graphene sheets were then transferred to IPA for deposition.

Size-selected AgNS inks were produced by diluting the purchased dispersion (Tokusen Nano, N300, $l_{NS} \approx 300 - 500$ nm) in DI water to a concentration of 100 mg mL$^{-1}$. This stock dispersion was then centrifuged at 28× $g$, 112× $g$, 447× $g$, and 4052× $g$ in 5 min steps. The sediment of the 112× $g$, 447× $g$ and 4052× $g$ steps was collected and redispersed in fresh DI water. The purchased AgNWs (Novarials, A60, ≈60 nm diameter) were diluted in IPA to a concentration of 0.5 mg mL$^{-1}$. The AgNW length was controlled by sonication-induced scission[87] of the wires using a sonic bath (Thermo Fisher Scientific, FB11201, 37 kHz). Size-selected inks were produced by sonication of the diluted dispersion for 0, 1, and 2 h respectively.

### Ink & nanosheet characterisation
Atomic force microscopy (Bruker Multimode 8, ScanAsyst mode, non-contact) was used to measure the nanosheet thickness and lateral dimensions in the size-selected graphene, $WS_2$ and AgNS inks. Measurements were performed in air under ambient conditions using Al-coated silicon cantilevers (OLTESPA-R3). The inks were diluted to optical densities <0.1 at 300 nm in IPA and a drop of dispersion (15 μL) was flash evaporated on pre-heated (175 °C) Si/SiO$_2$ wafers. After deposition, the wafers were rinsed with ≈15 mL of DI water and ≈15 mL of IPA and dried with compressed nitrogen. Typical image sizes ranged from 10 × 10 μm$^2$ for larger nanosheets to 3 × 3 μm$^2$ for small nanosheets at scan rates of 0.5 – 0.8 Hz with 1024 lines per image. Previously published length corrections were used to correct lateral dimensions from cantilever broadening[88]. Bright-field transmission electron microscopy (TEM) was performed using a JEOL 2100 LaB$_6$ system operating at 200 kV. Samples were diluted to a suitable optical density in IPA and drop cast onto holey carbon grids (Agar Scientific) on filter membranes to absorb excess solvent. The grids were left to dry in air and then placed overnight in a vacuum oven at 70 °C before measuring. UV-Vis optical spectroscopy (Perkin Elmer 1050 spectrophotometer) was performed to determine the graphene and $WS_2$ ink concentrations post-transfer[88,89]. Inks were diluted to a suitable optical density and extinction spectra were recorded in 1 nm increments using a 4 mm quartz cuvette. The concentration of the size-selected AgNS inks was found by vacuum filtration of a known volume of ink onto an alumina membrane (Whatman Anodisc, 0.02 μm pore size) and weighing. AgNW lengths were determined by drop casting 300 μL of ink, diluted to a concentration of 0.01 mg mL$^{-1}$, onto Au-coated Si/SiO$_2$ heated to 150 °C and measured from SEM images.

### Network deposition
Nanomaterial inks were spray coated using a Harder and Steenbeck Infinity airbrush attached to a computer-controlled Janome JR2300N mobile gantry. A N$_2$ back pressure of 45 psi, nozzle diameter of 0.4 mm and stand-off distance of 100 mm between the nozzle and substrate were used[90]. All traces were patterned using stainless-steel masks while the substrate was heated to 80 °C using a hotplate. The size-selected LPE graphene inks were sprayed at a concentration of 0.2 mg mL$^{-1}$ onto ultrasonically cleaned glass slides with prepatterned gold electrodes (Temescal FC2000 metal evaporation system) to facilitate electrical

measurements. Each trace was annealed overnight under vacuum at 80 °C to remove residual solvent. Electrochemically exfoliated graphene inks were deposited using identical parameters but were annealed for 2 h at 500 °C in a glovebox. $WS_2$ inks were spray coated at a concentration of 0.5 mg mL$^{-1}$. For the ITO:glass/$WS_2$/Au heterostack, a $WS_2$ ink was spray coated onto a purchased ITO:glass substrate (Ossila, 100 nm thick ITO, 2 μΩ m) and a 100 nm gold top electrode was deposited using a Temescal FC2000 system. The AgNS inks were deposited at a concentration of 10 mg mL$^{-1}$ onto Al$_2$O$_3$-coated PET (Mitsubishi Paper Mills) with the hotplate heated to 100 °C. AgNW inks were spray coated onto Si/SiO$_2$ substrates (MicroChemicals) at a concentration of 0.5 mg mL$^{-1}$. The purchased silver nanoparticle dispersion (Sigma Aldrich, <50 nm diameter, 30 – 35 wt% in methyl-triglycol) was diluted to a concentration of 20 mg mL$^{-1}$ and patterned using an aerosol jet printer (Optomec AJP300).

### Network characterisation
Electrical measurements were performed in ambient conditions using a Keithley 2612A source meter connected to a probe station. Two-terminal measurements were used to measure the resistivity of the printed LPE graphene networks in the transmission line configuration. Four-terminal measurements were used to measure the current-voltage characteristics of the printed graphene networks for the EE vs. LPE comparison. The thickness of the printed films was measured using a Bruker Dektak stylus profilometer (10 μm probe, 19.6 μN force). For all focused ion beam and scanning electron microscopy, the samples were directly mounted on an aluminum stub using a conductive carbon tab (Ted Pella) and grounded using silver paint (Ted Pella). Scanning electron microscopy of the network surfaces was performed using a Carl ZEISS Ultra Plus SEM at an accelerating voltage of 2 kV (5 mm working distance, 30 μm aperture, in-lens and SE2 detectors). FIB-SEM microscopy of network cross-sections was carried out using a dual beam Carl ZEISS Auriga system. All images were captured at a working distance of 5 mm with a 2 kV accelerating voltage and aperture size of 30 μm. FIB-SEM nanotomography was performed using ZEISS ATLAS 5 software (Version 5.3.3.31). All milling of network cross-sections was carried out using a 30 kV:600 pA gallium ion beam.

### Image processing and analysis
The generated image stacks for each network were aligned and reconstructed in 3D using Dragonfly (Version 2022.1.0.1231, Object Research Systems). Greyscale network cross-sections (SE2 detector) were segmented into their nanosheet and pore components using the Trainable WEKA Segmentation[45] plugin in FIJI[50]. Prior to applying the WEKA classifier, the pixel intensity in each network image was normalised. The image brightness and contrast were then adjusted globally across the entire stack using FIJI to ensure maximum contrast between the different phases. Measurements of the network tortuosity factor, specific surface area and porosity were performed using Taufactor[49]. Network porosity was also measured on a slice-by-slice basis using FIJI, which was again utilised to determine the pore size and circularity in each 2D cross-section. Isolated nanoplatelets in the reconstructed volumes were identified using a 3D Distance Transform Watershed in FIJI with a chessboard distance transform and dynamic setting of 2. To determine the Hermans orientation factor, these were converted into equivalent ellipsoids and the orientation of each was measured using the MorphoLibJ plugin[60] in FIJI. Connectivity analysis was performed using the Find Connected Regions plugin in FIJI. The nanosheet thickness and length in each network was measured from cross-sections using the Ridge Detection[58] plugin in FIJI. Each network was sampled at slice intervals that corresponded to 0.25 × $l_{NS}$. Reconstruction of the network surfaces was performed in FIJI. Network volumes were resliced in the $xz$-plane (top-down view of the network), where each slice has a thickness of 5 nm. The Z-stack Depth Colorcode

plugin in FIJI was then used to encode depth information into the slice-by-slice data. A different greyscale pixel intensity (from 255 to 0) was assigned to each sequential slice from the top surface of the network down. Each intensity value corresponds to a vertical height of 5 nm. Images of the depth-coded surfaces were generated using the Volume Viewer plugin and the surface roughness was determined from line-scans using the Plot Profile function in FIJI.

## Computer-generated slices

Computer generated intermediate slices were produced using Depth-Aware video frame INterpolation (DAIN)[85] trained on the Vimeo90k dataset without further fine tuning. DAIN's developers provide a Google Colaboratory notebook at https://github.com/baowenbo/DAIN where a sequence of frames can be uploaded. The number of new frames to be generated between each pair of consecutive frames in the original image stack is then selected. Here, every second frame from the original image stack was first removed and then replaced using DAIN. This allowed the generated images to be compared to the original ground-truth images.

## Statistics & reproducibility

The data are presented as means ± standard error (SE) in the mean. Combined uncertainty, where applicable, was calculated as the root sum of squares (RSS) of segmentation error and individual standard errors in the mean. No data were excluded from the analyses and the experiments were not randomised.

## Reporting summary

Further information on research design is available in the Nature Portfolio Reporting Summary linked to this article.

## Data availability

The data supporting the findings of this study are available from the corresponding author upon request. Source data are provided with this paper.

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

## Acknowledgements
J.N.C. greatly appreciates generous support from the European research council (FUTURE-PRINT), Graphene flagship core 3 (881603). L.D appreciates support from Science Foundation Ireland (SFI) (18/EPSRC-CDT/3581). L.G. appreciates support from SFI (12/RC/2278 – P2). J.M. acknowledges his Margarita Salas fellowship from the Spanish Ministry of Universities. L.J. appreciates support from SFI (URF/RI/191637). J.N.C, L.J., and S.S. appreciate support from SFI-funded centre AMBER (SFI/12/RC/2278). The authors have availed of the facilities of the SFI-funded advanced microscopy laboratory (AML) and additive research laboratory (ARL). We thank Dr. Megan Canavan and Mr. Clive Downing for valuable help and advice with the FIB-SEM-NT.

## Author contributions
J.N.C., C.G. and L.D. conceived and designed the experiments. C.G. and L.D. carried out the formal analysis. L.G. produced computer-generated intermediate images. C.G., L.D., J.M., E.Co., A.G.K., E.Ca. and S.L. produced samples for analysis. K.S. and J.M. performed AFM measurements. C.M. carried out exploratory experiments. 3D images were produced by C.G. J.N.C, L.J. and S.S. provided supervision. Funding was obtained by J.N.C. The manuscript was written and edited by C.G. and J.N.C.

## Competing interests
The authors declare no competing interests.
