## [Peer Review File · Nature Communications]

Quantitative analysis of printed nanostructured networks using high-resolution 3D FIB-SEM nanotomographyREVIEWER COMMENTS

Reviewer #1 (Remarks to the Author):

This is an interesting work. However, some issues are needed to address and major revision is required.

1. Several tools such as X-ray CT and Cryogenic electron tomography are already routinely used to characterize the morphology of electrochemical devices or materials. Please briefly describe the advantages of the High-resolution 3D FIB-SEM compared to other tools.
2. Moreover, this 3D reconstruction techniques are destructive and require invasive sample preparation methods, and accurate segmentation of the reconstruction data can be particularly challenging as material can be observed through the sample's porosity, which makes it difficult to identify exactly the extent of the pores of the sample. Please expound how to ensure the high accuracy and repeatability of the reconstruction data for conducting more quickly and accurately with ML techniques.
3. Please describe the temporal resolution of laboratory sources. Whether the High-resolution 3D FIB-SEM is capable of operando and in situ analyses for many structural dynamics in the range of minutes to hours.
4. Please explain whether the resolution of materials is affected by the charge effect.
5. Please state whether this method is applicable to all different sizes of 1D and 2D nanomaterials.
6. Please indicate whether the size of the nanomaterials will affect the identification of pores.
7. A CT machine was dedicated to metrology and industrial applications were introduced in 2005. Please illustrate whether the High-resolution 3D FIB-SEM can be applied to industrial applications and provide its resolution.
8. Please provide specifically the ML-orchestrated workflows which further enhance resolution in FIB-SEM-NT produced 3D volumes.

Reviewer #2 (Remarks to the Author):

The article entitled "Morphological Characterisation of Printed Nanostructured Networks using High-resolution 3D FIB-SEM Nanotomography" by Coleman and coworkers demonstrates the application of 3D FIB-SEM Nanotomography for characterizing the morphology of nanostructured networks. The authors investigated the relationship between nanosheet/nanowire size and network structure for versatile nanomaterials-printed networks and studied their electrochemical properties.

While the authors have applied the technology for high-resolution structural characterization, achieving a voxel size of 5 nm × 5 nm × 15 nm, the novelty of the work is really limited, and not appropriate to be published in Nature Commun., but rather in a more specialized engineering journal.

1. Although this study shows some high-resolution structural images of Printed Nanostructured Networks, the FIB-nanotomography method itself is not new, as evidenced by previous publications (Catalysis Today, 405-406,1, 2022; Ultramicroscopy, 163, 38-47, 2016; Journal of Microscopy, 281: 76-86, 2020).
2. While the paper provides insights into the influence of size on morphology, porosity, and tortuosity,

the authors primarily focus on the technological advancements rather than the physiochemical property investigation of the nanostructured network.

3. A more comprehensive analysis and discussion should and could be achieved by providing a comparison of different imaging technologies, addressing factors such as resolution limit (voxel size) and sample destruction (which is the drawback of FIB-SEM). This would help readers better understand the advantages and limitations of the employed FIB-SEM Nanotomography method in comparison to other techniques.

Reviewer #3 (Remarks to the Author):

Significant results were shown in the article about characterization with 3D imaging, the study of the pore sizes, orientation (size dependence) and its versatility in 3D imaging different types of 2D nanomaterials. I had some minor questions/comments. It is related to and improved from current literature. There are enough data to reproduce the work, and the methodology is sound, meeting the expectation of the field.

The work supports the claims, and I had a few comments:

1. Write the full name term for FIB-SEM at the first abbreviation (page 1)
2. For reference 22 - Do you have more examples? From this paper, the only sample preparation is removing water and impurities and packing the samples. This doesn't seem like "considerable" sample preparation for me.
3. Are there limitations for the Dragonfly for the image stack alignment and 3D reconstruction? (page 3 and SI)
4. Does the spray control the placement/distribution of graphene? If so, how would that affect the diffusive flux (page 4)
5. Can the spray also affect the porosity study shown in Fig 3?
6. Figure 4D - it is hard to see Figure 4D

Here we respond to the reviewers' comments for the manuscript "Morphological Characterisation of Printed Nanostructured Networks using High-resolution 3D FIB-SEM Nanotomography". We believe we have addressed all comments successfully as described below.

We apologize in advance as some of the responses are quite long. However, given the nature of some of the comments, we felt that comprehensive responses were best.

Reviewer 1

This is an interesting work. However, some issues are needed to address and major revision is required.

We thank the reviewer for their evaluation and comments on our work. We have edited the manuscript in line with their suggestions as detailed below.

Point 1: Several tools such as X-ray CT and Cryogenic electron tomography are already routinely used to characterize the morphology of electrochemical devices or materials. Please briefly describe the advantages of the High-resolution 3D FIB-SEM compared to other tools.

Response:

There are a range of techniques that are used for tomography at present, each with distinct advantages and limitations for characterising different systems. We address this in the context of printed nanosheet/nanowire devices for X-ray CT and cryogenic electron tomography here.

X-ray CT has been widely utilised for microstructural analysis and generally facilitates tomography on larger volumes than FIB-SEM NT (FIB-SEM NT volumes are $\sim(10 \mu\text{m})^3$). However, there is a trade-off between the sampled volume and the spatial resolution (Materials Today, **10**, 26-34, 2007. DOI). For example, while micro CT can be used reconstruct volumes greater than $(100 \mu\text{m})^3$ in size, it is limited to voxel sizes $> 100 \text{ nm}$ (Nature Reviews Methods Primers, **1**, 2021. DOI). Nano CT can image sample volumes up to $(\sim 100 \mu\text{m})^3$ but is limited to voxel sizes on the order of tens of nanometres (ACS Nano, **15**: 15342-15353, 2021. DOI). In both cases, such a spatial resolution is too low to fully characterise the nanostructure of printed nanosheet/nanowire networks where the smallest nanowire/nanosheet dimension is generally $< 50 \text{ nm}$, and in some cases, $\ll 50 \text{ nm}$. These resolution limitations mean there are few reports of nano CT on networks of nanosheets/nanowires. A recent paper reconstructed vacuum filtered films of Mxene nanosheets ($\sim 2 \text{ nm}$ thick, $\sim 5.4 \mu\text{m}$ long) using a voxel size of 32 nm to measure porosity and pore size distributions (Science, **374**: 96-99, 2021. DOI). However, the authors note that the measured porosity deviates by a factor of ~ 2.8 from density measurement values as "*very small voids, with a voxel size of dozens of nanometres, could not be observed*" at this resolution. A second paper reported imaging of 3D printed microcapacitors comprised of graphene platelets ($\sim 3 \text{ nm}$ thick, $\sim 600 \text{ nm}$ long) using voxel sizes of 62 nm and 126 nm (ACS Nano, **15**: 15342-15353, 2021. DOI). These papers have voxel volumes are a factor of ~ 90 , 640 and 5000 larger than the voxel volume of 375 nm^3 ($5 \times 5 \times 15 \text{ nm}$) utilised in this work. Similarly, a previous work used nano CT to image a composite containing silver nanowires (AgNW) with a voxel size of 65 nm (Advanced Theory and Simulations, **3**: 2020. DOI). However, the AgNWs had a diameter 300 nm which is much larger than that used in most applications. In comparison, the printed networks of AgNWs characterised in this work that have a diameter of 55 nm .

To highlight the resolution advantage of FIB-SEM NT for accurate reconstruction of printed nanosheet networks we have added a comparative figure to the SI (Fig. S15). Fig. S15A shows an image obtained from a vacuum filtered silver nanosheet film using a lab-based micro CT microscope (Nikon XTH 225, voxel size = $4 \times 4 \times 4 \mu\text{m}$). The silver nanosheets (AgNS) were vacuum filtered to ensure the film was thick enough to be imaged (thickness = $80 \mu\text{m}$) and because the contrast between the AgNS network and the polyester filter membrane was sufficient to allow visualisation of the network. The voxel size of $4 \mu\text{m}$ is much larger than both the nanosheet length ($l_{\text{NS}} = 630 \text{ nm}$) and thickness ($t_{\text{NS}} = 71 \text{ nm}$), so no information on the internal structure of the network is obtained. A spray coated network of the same AgNS reconstructed using FIB-SEM NT is shown in Fig. S15B. Here, the enhanced resolution (voxel size = $5 \times 5 \times 15 \text{ nm}$) allows individual nanosheets to be resolved, as well their orientation and dimensions.

Fig. S15: Network structure for different imaging techniques and pixel/voxel sizes. (A) Image of a vacuum filtered silver nanosheet (AgNS) network on a filter membrane captured using an X-ray micro CT scanner with a voxel size of $4 \times 4 \times 4 \mu\text{m}$. (B) A printed network of the same AgNS reconstructed using FIB-SEM NT with a voxel size of $5 \times 5 \times 15 \text{ nm}$. (C-D) FIB-SEM cross-sections of a printed graphene network captured using a pixel size of (C) 50 nm and (D) 5 nm. (E-F) Reconstructed volumes of the same printed graphene network a voxel size of (E) $50 \times 50 \times 50 \text{ nm}$ and (F) $5 \times 5 \times 15 \text{ nm}$. The pixel and voxel sizes used in (C) and (E) are representative of the x-ray nano CT technique.

A significant drawback to X-ray nano CT is that the technique is highly specialised and not widely accessible. Thus, we captured FIB-SEM cross-sections of the same printed graphene network at a pixel size of 50 nm, as a proxy for the typical resolution achievable using nano CT (Fig. S15C), and at the pixel size used in this work of 5 nm (Fig. S15D). The loss of information in Fig. S15C is striking when compared to the image in Fig. S15D, where the pixel size is larger than the nanosheet thickness in many cases. This causes the interphase boundaries (nanosheet/pore/substrate/platinum) to become blurred, which translates to a very poor reconstruction of the network structure in Fig. S15E. Such volumes would clearly not offer accurate morphological information and highlights the need for the spatial resolution FIB-SEM NT can provide, as shown for the same network in Fig. S15F. For example, it has been shown that increasing the resolution of an imaging technique allows smaller pores to be detected and increases the measured pore connectivity and specific surface area (Journal of The Electrochemical Society, **159**: A1023-A1027, 2012. DOI).

Cryogenic electron tomography (3D-TEM) has a superior spatial resolution to FIB-SEM NT, allowing for the visualisation of individual carbon nanotubes in polymer nanocomposites (Journal of Materials Research, **29**: 1817-1823, 2014. DOI), or each individual atom within a region of interest (Nature, **483**, 444-447, 2012. DOI). However, this is achieved at the cost of imaging a much smaller volume. Samples for 3D-TEM must be electron transparent, meaning the volumes to be analysed must be less than $\sim 500 \text{ nm}$ thick (Nat. Methods, **20**, 499-511, 2023. DOI). This can require laborious sample preparation, generally using a microtome (RSC Advances, **4**: 9300-9307, 2014. DOI) or a focused ion beam microscope (Journal of Structural Biology, **180**: 318-326, 2012. DOI) to prepare ultra-thin sections of the appropriate dimensions. Crucially, this thickness requirement for electron transparency, as well as the required tilting process for 3D-TEM, limits the imaged volumes to be on the order of $\sim 1 \mu\text{m}^3$ in size (The Journal of Physical Chemistry C, **115**: 14236-14243, 2011. DOI).

Such small network volumes would not be expected to be representative of a complete nanostructured network or device – many of the nanosheets in this work are $> 500 \text{ nm}$ long, while the AgNW length in the networks analysed range between $4 - 23 \mu\text{m}$. Thus, meaningful morphological properties that we calculate in the main text including network porosity, pore shape/size, nanosheet/pore tortuosity and specific surface area cannot be determined accurately using 3D-TEM. For example, we show in the main text (Fig. 2B) that the

measured porosity from a single network cross-section ($\sim 0.53 \mu\text{m}^3$) can vary by as much as $\pm 10\%$ from the average value across the entire network volume. Properties such as pore connectivity and tortuosity are far more severely affected by analysis of small volumes. We have added a figure to the SI (Fig. S16), showing reduced volume analysis of the tortuosity factor, κ , of the pore volume for a printed graphene network ($l_{\text{NS}} = 298 \text{ nm}$).

Fig. S16: Reduced volume analysis for the network tortuosity factor. Plot of the measured the tortuosity factor, κ , of the pore volume in the in-plane z -direction (the slicing direction) for a network of printed LPE graphene nanosheets ($l_{\text{NS}} = 298 \text{ nm}$) as a function of the sampled network volume. The red line is a fit to a single exponential decay. κ_0 is the fitted plateau value for the tortuosity factor and κ_{Exp} is the measured value for the full volume imaged using FIB-SEM NT.

Here, we took reduced volumes from the network and measured the tortuosity factor, κ , of the pore volume in the in-plane z -direction (the slicing direction). Fig. S16 shows that the measured tortuosity factor is overestimated for smaller network volumes, likely due to the reconstructed volume being too small for isolated pores to coalesce. However, we see κ decay with increasing sample volume as the discrete pores become connected and that κ stabilises to a plateau for values greater than $\sim 200 \mu\text{m}^3$ (*cf.* sampled volumes of $1 \mu\text{m}^3$ for 3D TEM). This highlights that while 3D TEM may offer higher resolution than FIB-SEM NT, the sampled volumes are likely too small to fully describe the morphology of printed nanosheet/nanowire networks and their devices. The data in Fig. S16 also demonstrates that while nano CT can probe larger sample volumes, the achievable volumes of $10^2 - 10^4 \mu\text{m}^3$ using FIB-SEM NT are large enough to be representative of the networks studied here.

We have added **Figs. S15 & S16 to the SI**, as well as a more detailed comparison between FIB-SEM NT and other tomographic techniques to the manuscript.

Main text, page 2:

“Established 3D-imaging techniques such as X-ray computed tomography (X-ray CT) or electron tomography (3D TEM) are routinely used to characterise samples for metrological²⁵, biological²⁶ and materials science applications^{27,28}. However, these techniques can require non-trivial sample preparation^{29,30} and there is a trade-off between the sampled volume and spatial resolution. X-ray nano CT can probe larger sample volumes than FIB-SEM NT but is limited by voxel sizes on the order of tens of nanometres³¹. This has been shown to be inadequate to characterise the morphology of nanosheet networks as pores or nanosheets less

than ~50 nm in size cannot be resolved³². Insufficient spatial resolution can also cause interphase boundaries to become blurred, which has been linked to underestimations in the measured pore connectivity and specific surface area in battery electrodes³³. Alternatively, electron tomography can offer sub-nanometre resolutions³⁴ for electron transparent samples (< 500 nm thick)²⁶. However, this is at the cost of drastically reduced sample volumes of ~ 1 μm^3 ³⁵. Volumes this size are not expected to be representative of an entire printed nanosheet network where the constituent nanosheets are often > 500 nm in length, or printed heterostacks with thicknesses > 1 μm . FIB-SEM nanotomography (FIB-SEM-NT) effectively bridges the gap between these tomographic techniques by offering spatial resolutions of a few nanometres over representative sample volumes of $10^2 - 10^4 \mu\text{m}^3$ ⁶⁸.”

Main text, page 4

“Crucially, FIB-SEM-NT facilitates analysis of these 2D networks over representative volumes and at a resolution that preserves the discrete nanosheet and pore components (see SI).”

Point 2: Moreover, this 3D reconstruction techniques are destructive and require invasive sample preparation methods, and accurate segmentation of the reconstruction data can be particularly challenging as material can be observed through the sample’s porosity, which makes it difficult to identify exactly the extent of the pores of the sample. Please expound how to ensure the high accuracy and repeatability of the reconstruction data for conducting more quickly and accurately with ML techniques.

Response:

We thank the reviewer for this comment. Regarding sample preparation for FIB-SEM NT, we contend that this technique arguably requires less invasive sample preparation than other tomographic techniques, such as X-ray nano CT and 3D TEM. X-ray nano CT can require nontrivial sample preparation including mechanical sectioning, laser cutting, FIB milling or a combination of all these techniques to produce a sample thin enough to be imaged (Journal of Microscopy, **267**: 384-396, 2017. DOI). Similarly, sample lamellae for 3D TEM must be laboriously thinned down to electron transparent thicknesses of < 500 nm using a microtome (RSC Advances, **4**: 9300-9307, 2014. DOI) or a focused ion beam microscope (Journal of Structural Biology, **180**: 318-326, 2012. DOI). Following this, the ultra-thin samples must then be carefully transferred to the X-ray CT scanner/TEM for analysis. Thus, both techniques can require extended preparation steps inside a FIB microscope before even beginning a measurement. For FIB-SEM NT the sample/substrate need only be mounted on a SEM stub and appropriately grounded using conductive silver paint. Many of the samples in this work were printed on PET substrates or glass microscope slides (Fig. S3B), which were placed directly into the FIB-SEM for analysis. We also discuss how the sample requirements for mercury intrusion porosimetry (MIP) and BET N₂ analysis are incompatible with printed nanosheet/nanowire networks in response to **Reviewer 3 (Point 2)**. We have clarified the above in the main text.

Introduction, page 2

“However, these techniques can require non-trivial sample preparation^{29,30} and there is a trade-off between the sampled volume and spatial resolution.”

Materials and Methods, page 14

“For all focused ion beam and electron microscopy the samples were directly mounted on a SEM stub using a conductive carbon tab (Ted Pella) and grounded using silver paint (Ted Pella)”

FIB-SEM NT is indeed a destructive tomographic technique. After a network cross-section is imaged using the SEM, the focused ion beam then sputters a ~15 nm slice from the network to reveal the next cross-section. This is the most significant drawback of FIB-SEM NT and can preclude *in-situ* measurements of dynamic processes, for example, the microstructural evolution of battery materials during cycling. As discussed above, FIB-SEM NT doesn’t require macroscopic destruction of the sample (the effective destroyed sample area in

this work was $\sim 50 \times 50 \mu\text{m}$). Thus, it is possible to perform longitudinal studies on different regions of the same sample/device at different stages of its evolution using FIB-SEM NT. We have clarified that FIB-SEM NT is a destructive tomographic technique in the main text (Page 3):

“To achieve this (Fig. 1C), ~ 800 network cross-sections are sequentially milled using the FIB and imaged using the SEM (see Supplementary Information, SI). Each network slice has an in-plane pixel size of 5 nm and average thickness of ~ 15 nm, giving a voxel size of $\sim 375 \text{ nm}^3$. While this is a destructive technique that removes material from the network, it can confer voxel sizes 10 – 1000 times smaller than conventional X-ray CT scanners^{43,44}.”

The shine-through effect, which causes material to be seen through a sample's porosity, is well-reported issue in FIB-SEM NT (Ultramicroscopy, **219**: 2020. DOI). In this work we applied numerous measures to minimise this. Each sample region of interest was surrounded by a large viewing trench and two side trenches to reduce shadowing and redeposition effects (Fig. S11A). All cross-sectional images were captured using the SE2 detector at a low accelerating voltage of 2 kV. This served to reduce the imaging depth of field into pores (Microscopy and Microanalysis, **3**: 1241-1242, 1997. DOI), alleviate charging effects (Scanning Electron Microscopy and X-Ray Microanalysis, 2018. DOI) and increased topographic contrast between the nanosheets and pores when compared to the Inlens detector (Catalysis Today, **405-406**: 2-13, 2022. DOI). The image brightness and contrast were then adjusted globally across the entire image stack using *FIJI* to ensure maximum contrast between the different phases. We have added a figure to demonstrate the above in the SI (Fig. S12).

Fig. S12: Detector choice and image pre-processing. High magnification cross-sectional SEM images of the same region of a printed silver nanosheet network. Images (A) and (B) were captured using the Inlens and SE2 detectors respectively. Image (C) was captured using the SE2 detector and the brightness/contrast was adjusted to reduce the shine-through effect and improve classification of the nanosheets and pores.

This led to improved image classification by ensuring that only nanosheets/pores at the cross-section face were considered in the segmentation process (Fig. S12C). The images were then classified using *trainable WEKA segmentation*, as detailed in the SI (Section 3 – Image Segmentation). Regions of each image were first manually assigned as either “pore” or “nanosheet” to provide training data for the classifier (Fig 1D, main text). We then segmented the image stacks using an array of classification features including edge-detection and textural features to ensure sharp boundaries between the nanosheets/pores and reduce imaging artefacts. These are described in detail in the SI (page 13). The classification algorithm used was a multi-threaded random forest classifier with 200 uncorrelated decision trees and two random features per node (Machine Learning, **45**, 5-32, 2001. DOI). Such classifiers have shown comparable performance to manual segmentation by an expert for porous polymer networks (Journal of Microscopy, **281**, 76-86, 2020. DOI). The performance of the classification process was assessed through the out-of-bag (OOB) error, which is a measure of how well decision trees in the model perform in classifying subsets of data that they were not directly trained on. For networks in this work the classification algorithm was trained until the OOB error was below 2%. We have placed additional information and citations in the *Image Segmentation* section of the SI, and have clarified the steps taken to improve the image classification process in the main text (Page 3):

“To enable quantitative analysis, each image in the stack was classified into its pore and nanosheet components using trainable WEKA segmentation⁴⁵ (see SI). All images were captured at an accelerating voltage of 2 kV using the SE2 detector to alleviate shine-through effects⁴⁶. This ensured that only nanosheets and pores at the cross-section face were considered for classification (see SI). As shown in Fig. 1D, regions of each slice were first manually assigned as either nanosheet or pore, providing training data for the classifier. Each slice was then segmented into these components on a pixel-by-pixel basis using a random forest classifier⁴⁷ to create a binary image.”

Point 3: Please describe the temporal resolution of laboratory sources. Whether the High-resolution 3D FIB-SEM is capable of *operando* and *in situ* analyses for many structural dynamics in the range of minutes to hours.

Response:

To the best of the authors knowledge FIB-SEM NT has not been utilised for *operando* and *in-situ* analysis of structural dynamics, as is possible with synchrotron X-ray nano CT using specially designed setups e.g. electrochemical cells (Angewandte Chemie International Edition, **53**: 4460-4464, 2014. DOI). This arises from the destructive nature of FIB-SEM NT as well as sample and equipment constraints. To protect the sample from ion beam damage and reduce imaging artefacts it is necessary to deposit a protective platinum pad over the sample region of interest (Figs. S10 & S11). Dynamic processes such as battery electrode swelling upon lithiation could disrupt/destroy this pad and crucially, the 3D tracking marks encased within. Furthermore, samples within specially designed apparatuses, such as the electrochemical cell designed to mimic a battery electrode above, cannot be imaged *in-situ* due to the destructive nature of FIB-SEM NT. Finally, FIB-SEM NT works via a slice-and-scan process, where a slice can take seconds to image and mill, which is likely too slow for *operando* analyses of dynamic processes. However, we mention above that FIB-SEM NT doesn't require macroscopic destruction of the sample. Thus, it is possible to perform longitudinal studies on different regions of a sample/device at different stages of its evolution, for example pre and post mechanical calendaring. Similarly, numerous reports have used a FIB-SEM to compare the morphology of battery electrodes pre- and post-cycling using individual cross-sections (Journal of Power Sources, **306**: 300-308, 2016. DOI).

Point 4: Please explain whether the resolution of materials is affected by the charge effect.

Response:

Numerous steps were taken to ensure that sample charging was negligible for the printed graphene, WS₂, silver nanosheet (AgNS) and silver nanowire (AgNW) networks in the main text. All samples were mounted on the SEM stub using conductive carbon tabs. A conductive grounding path was then added to each sample using silver paint. An electron beam energy of 2 keV was chosen to produce as little charging as possible while sustaining a good detector signal. FIB-SEM NT using low beam energies in the range 0 – 2 keV has been demonstrated on soft, porous, and poorly conducting materials (Microscopy and Microanalysis, **26**: 837-845, 2020. DOI). Furthermore, the slice acquisition time (to mill the cross-section face and image it) was minimised to < 30s by increasing the scan rate (dwell time of 0.7 μ s) and line averaging. Finally, the SE2 detector was chosen to reduce sample charging when compared to the In-lens detector. The materials in this work were all relatively conductive, with resistances ranging from 10 – 10⁶ Ω for the AgNS and WS₂ samples respectively. For FIB-SEM NT of insulating networks it is possible to alleviate charging by injecting carbon or nitrogen gas in the vicinity of the sample before imaging each slice (Microscopy and Microanalysis, **15**: 332-333, 2009. DOI). However, this was not required for this work. We have added further information about sample grounding and the imaging conditions to the manuscript.

Main text, page 3

“All images were captured at an accelerating voltage of 2 kV using the SE2 detector to alleviate sample charging and shine-through effects⁴⁶.”

Main text, page 14

“For all focused ion beam and electron microscopy the samples were directly mounted on an aluminium stub using a conductive carbon tab (Ted Pella) and grounded using silver paint (Ted Pella).”

Point 5: Please state whether this method is applicable to all different sizes of 1D and 2D nanomaterials.

Response:

The range of nanomaterial sizes that can be examined using FIB-SEM NT is limited by the smallest pixel size achievable in the SEM at the lower end (~ 5 nm in this work), and the maximum sampled volume ($\sim 10^4 \mu\text{m}^3$) at the higher end. A general rule of thumb in tomography is that an object should take up at least 3 voxels to be positively identified (Nat Rev Methods Primers, 1, 2021. DOI). This precludes imaging of the smallest nanomaterials (dimensions < 5 nm), such as monolayered nanosheets or isolated and small bundles of single-walled carbon nanotubes. However, it may be possible to assess the structure of their aggregated and bundled networks using FIB-SEM NT. The maximum sample areas of $\sim 50 \times 50 \mu\text{m}$ should be sufficient for networks of most 0D/1D/2D nanomaterials. Thus, we believe this method to be broadly applicable to most sizes of 0D, 1D and 2D materials. We have added a table summarising the range of nanomaterial dimensions that were reconstructed and analysed in this work (Table S1) and clarified this in the main text (page 9).

“Significantly, the data in Fig. 5 highlights the applicability of FIB-SEM-NT to characterise networks of 1D and 2D nanomaterials with feature sizes that range from a few nanometres to tens of microns (Table S1).”

	Nanomaterial	Length	Thickness	
0D	Silver Nanoparticles	50 nm	50 nm	Fig. 6B
1D	Silver Nanowires	4.3 – 22.9 μm	55 nm	Fig. 5C, F
2D	LPE Graphene	80 – 1087 nm	2.4 – 38 nm	Figs. 2-4
	EE Graphene	2.3 μm	3.7 nm	Fig. 6A
	WS ₂	127 – 1013 nm	17 - 45 nm	Fig. 5A,D
	Silver Nanosheets	160 – 900 nm	48 – 100 nm	Fig. 5 B,E

Table S1: Dimensions of the 0D, 1D and 2D nanomaterials that were reconstructed using FIB-SEM NT in this work.

Point 6: Please indicate whether the size of the nanomaterials will affect the identification of pores.

Response:

As we discuss in our response to **Point 5**, a guideline to positive identification of a phase (pores in this case) is that it takes up 3 pixels in 2D or 3 voxels in 3D. This confers a minimum measurable pore size of ~ 9 nm. For the absolute smallest 1D nanomaterials such as SWNTs, resolving the intra-bundle voids (~ 1 nm) is not possible using FIB-SEM NT. Similarly, for extremely well-aligned networks of ultra-thin electrochemically exfoliated (EE) MoS₂ nanosheets ($t_{\text{NS}} = 3.6$ nm), the plane-to-plane contacts can have an interlayer spacing of 6.1 Å (Nature, 562: 254-258, DOI). Here, techniques such as 2D cross-sectional TEM are required to image the intersheet voids. However, it is still possible to measure the mesoporosity (pore sizes > 5 nm) of highly-aligned networks of EE nanosheets using FIB-SEM (ACS Nano 2023, 17: 2912–2922, 2023. DOI). The examples above involve electrochemically exfoliated nanosheets produced using state-of-the-art protocols and were deposited via spin-coating and the Langmuir-Schaefer technique, respectively. For high-throughput applications such as low-cost and semi-disposable electronics, printing remains the preferred technique due to its scalability and speed (Nature Reviews Materials, 7: 717-735, 2022. DOI). We reconstruct a spray-coated

network of highly-aligned EE graphene nanosheets in the main text (Fig. 6A) and measure its surface roughness, tortuosity factor and mesoporosity. Furthermore, the high-yields and universal applicability of liquid phase exfoliation means that printed networks of LPE nanosheets are particularly relevant to these applications (Advanced Materials, **28**: 6136-6166, 2016. DOI). As we show in the main text (Fig. 3B-D), networks comprised of smaller LPE nanosheets are more densely packed and contain smaller pores. Here, the nanosheet lengths ranged from $l_{NS} = 1087$ nm ($t_{NS} = 38$ nm) to $l_{NS} = 80$ nm ($t_{NS} = 2.4$ nm) but in all cases the measured characteristic pore size (given by the average pore cross-sectional area) was considerably larger than the minimum measurable value. Taken together, this suggests that FIB-SEM NT is generally applicable to measure pores in printed LPE nanosheet networks of different sizes, while offering valuable information on the mesoporosity of highly-aligned networks where the smallest pores may be unresolvable in a SEM. We have made the following additions to the manuscript:

Main text, page 6:

“Taken together, the data in Fig. 3C-E suggests that changing the nanosheet size offers a simple means to tailor the network porosity for a target application. FIB-SEM-NT can be used to inform this by measuring pore sizes that span from a few nanometres to microns.”

Main text, page 9:

“Spin-coated networks of conformally tiled and high aspect ratio EE nanosheets have demonstrated basal plane separations < 1 nm, which are only resolvable using cross-sectional TEM¹⁵. However, it is still possible to characterise the nanostructure and mesoporosity (pore sizes > 5 nm) of such highly-aligned networks using the spatial resolutions afforded by FIB-SEM-NT².”

Point 7: A CT machine was dedicated to metrology and industrial applications were introduced in 2005. Please illustrate whether the High-resolution 3D FIB-SEM can be applied to industrial applications and provide its resolution.

Response:

FIB-SEM tomography has been utilised to characterise materials and devices in a wide variety of fields. We have compiled a table with example application areas, voxel sizes and sampled volumes below. Previous demonstrations of the FIB-SEM NT have been added to the **main text (Page 2)**.

“FIB-SEM nanotomography (FIB-SEM-NT) effectively bridges the gap between these tomographic techniques by offering spatial resolutions of a few nanometres over representative sample volumes of $10^2 - 10^4 \mu\text{m}^3$ ³⁸. This has been demonstrated through high-resolution reconstructions of oil shales³⁷, drug release coating³⁸, fuel cells³⁹ and commercial battery electrodes⁴⁰.”

Application	Voxel Size	Sampled Volume	Citation
Oil Shale Pyrolysis	$15 \times 18 \times 10$ nm	$3803 \mu\text{m}^3$	DOI
Drug Release Coatings	$10 \times 10 \times 50$ nm	$6000 \mu\text{m}^3$	DOI
Commercial LiCoO ₂ cathode	$31 \times 31 \times 150$ nm	$35000 \mu\text{m}^3$	DOI
Dental Veneers	$10 \times 10 \times 10$ nm	$87 \mu\text{m}^3$	DOI
Sea Urchin Embryos & Zebra Fish Tails	$10 \times 10 \times 20$ nm $20 \times 20 \times 40$ nm	$2640 \mu\text{m}^3$ $23030 \mu\text{m}^3$	DOI
Pyrocarbon-Silicon Carbide Interfaces for Nuclear Fuels	$26 \times 33 \times 40$ nm	$1648 \mu\text{m}^3$	DOI
Solid Oxide Fuel Cells	$10 \times 10 \times 10$ nm	$252 \mu\text{m}^3$	DOI

Point 8: Please provide specifically the ML-orchestrated workflows which further enhance resolution in FIB-SEM-NT produced 3D volumes.

Response:

We have added further details on the machine learning interpolation process to the **SI (Section 12)**.

“One of the video frame interpolation algorithms that can be used for the resolution enhancement of FIB-SEM-NT-produced 3D volume is Depth-Aware video frame INterpolation (DAIN)³⁶. It is made of two main components: a depth estimation network, which investigates quantities such as the distance between items in the frame, and a flow estimation network, used to determine the motion of pixels between consecutive frames. DAIN’s developers provide a ready-to-use Google Collaboratory notebook (<https://github.com/baowenbo/DAIN>) that can be easily accessed and run, with the possibility to use free GPUs provided by Google to accelerate the generation process. It should be noted that the model is already trained on a large dataset, namely the Vimeo90K dataset³⁷, and there is no need for fine-tuning. The notebook is configured to accept a video as input and to convert it into a sequence of frames. For this application, the notebook can be easily altered to accept frames (images in a stack) as input directly. After this small adjustment, the user only needs to specify the data path of the input folder, containing the experimentally captured frames, the data path of the output folder, where the new dataset will be saved, and finally the number of new frames that the model would need to generate between each couple of consecutive frames. For example, to homogenise the anisotropic $5 \times 5 \times 15$ nm voxels used in this work, DAIN could be used to generate 2 intermediate frames between each slice in the milling (z) direction to produce voxels with $5 \times 5 \times 5$ nm dimensions. The outcome is a folder containing a sequence of ordered frames, which form a volume of higher resolution compared to the original experimentally captured volume.”

We have also clarified the DAIN workflow in the **main text (page 11)**:

“This strategy can improve the resolution along the milling direction of FIB-SEM-NT data by introducing intermediate slices between the imaged cross-sections. To test this approach, every second network cross-section was removed from a printed LPE graphene image stack, effectively doubling the slice thickness from 15 nm to 30 nm. The removed frames were then replaced with images generated by DAIN to allow for a direct comparison. This doubled the resolution in the milling direction and restored the original slice thickness of 15 nm using generated images. These can then be compared to the removed ground-truth frames, as displayed in Fig. 7A-C. We find extremely good agreement, showing that neural-network based approaches can be used to further enhance resolution in FIB-SEM-NT generated 3D images.”

Reviewer 2

The article entitled “Morphological Characterisation of Printed Nanostructured Networks using High-resolution 3D FIB-SEM Nanotomography” by Coleman and coworkers demonstrates the application of 3D FIB-SEM Nanotomography for characterizing the morphology of nanostructured networks. The authors investigated the relationship between nanosheet/nanowire size and network structure for versatile nanomaterials-printed networks and studied their electrochemical properties.

While the authors have applied the technology for high-resolution structural characterization, achieving a voxel size of $5 \text{ nm} \times 5 \text{ nm} \times 15 \text{ nm}$, the novelty of the work is really limited, and not appropriate to be published in Nature Commun., but rather in a more specialized engineering journal.

We thank the reviewer for their appreciation and critical evaluation of our manuscript. We will aim to address their concerns about the novelty of this work in detail across the various points that they raise while offering a general summary here.

Nanomaterials are currently at the absolute cutting edge of materials science. One of the most important classes of nanostructured materials are solution-processed films or networks which are used in a range of applications from electronics (Nature Electronics, **6**, 2023. DOI) to energy storage (Science, **366**, 2019. DOI). As we show below, the properties of such systems depend critically on morphology. There is currently no practically useful method to quantify morphology in such systems. Thus, we must respectfully disagree with the comment that “the novelty of the work is really limited”.

High impact papers on solution processed networks of 0D (e.g. Nature Communications, **11**, 2020. DOI), 1D (e.g. Nature Materials, **16**, 2017. DOI), and 2D (e.g. Nature Communications, **14**: 278, 2023. DOI) nanomaterials have reported on application areas including printed electronics, sensing, catalysis, and energy storage. However, while many of these devices seek to capitalise on the superlative properties of their constituent nanomaterials, the physicochemical properties of their networks are well-reported to be limited by their morphology (Nature Electronics, **2**, 378-388, 2019. DOI) (Nature Reviews Materials, **7**: 217-234, 2021. DOI). We discuss one example of this in the context of thin-film nanosheet transistors in the main text (page 1):

“Printed 2D networks tend to consist of porous, disordered arrays of nanosheets with variable degrees of connectivity, alignment, and inter-sheet coupling. These morphological factors have been shown to heavily influence carrier mobility in nanosheet devices, where printed networks of poorly-aligned MoS₂ nanosheets¹⁴ demonstrate values of $\sim 0.1 \text{ cm}^2 \cdot \text{V}^{-1} \cdot \text{s}^{-1}$, but spin-coated networks of conformally tiled nanosheets¹⁵ exhibit mobilities of $\sim 10 \text{ cm}^2 \cdot \text{V}^{-1} \cdot \text{s}^{-1}$.”

Similarly, printed nanosheet capacitors require continuous dielectric layers. However, the lowest network thickness for a pinhole-free film of boron nitride nanosheets is seen to vary across 3 orders of magnitude depending on the deposition technique used (Nature Reviews Materials, **7**: 217-234, 2021. DOI). In sensing applications, the gauge factor of printed strain sensors is predominantly controlled by network structure (ACS Applied Materials & Interfaces, **14**: 7141-7151, 2022. DOI). Electrode morphology also plays a dominant role in nanosheet-based energy storage applications, where the network architecture is known to heavily influence rate performance in battery electrodes (Nature Communications, **10**, 2019. DOI). While such examples highlight the role of morphology in device performance, analysis is often limited to qualitative discussions of isolated SEM or FIB-SEM images (Nature Communications, **11**, 2020. DOI). Quantitative characterisation of the internal structure of these systems remains largely unreported due to resolution limitations or sample constraints, which represents a significant gap in our knowledge.

In this work, we present 3D reconstructions of printed nanoparticle, nanosheet and nanowire networks at an unprecedented resolution of $5 \times 5 \times 15 \text{ nm}$. Only two papers have reported FIB-SEM NT of a nanosheet network to this point, neither of which were printed. The first utilised a voxel size of $\sim 30 \text{ nm}$ to measure porosity and pore volume distributions of vacuum filtered Mxene films (Science, **374**: 96-99, 2021. DOI).

Crucially, the authors note that the measured porosity deviates by a factor of ~2.8 from density measurement values as “*very small voids, with a voxel size of dozens of nanometres, could not be observed*” at this resolution. The second paper reports FIB-SEM NT of a slurry-cast Fe₃O₄ electrode (voxel size ~ 11 × 11 × 25 nm) for visual assessment of the electrode structure (Nano Letters, **22**: 6700-6708, 2022. DOI). The morphology of printed networks of graphene, WS₂ and silver nanosheets, as well as silver nanowires has not been reconstructed or quantitatively analysed at this resolution using any technique. Furthermore, this work is the first to reconstruct printed vertical heterostacks or devices comprised of 0D or 2D materials in 3D, whereby discrete layers are isolated for analysis as well as the interfacial properties between them including surface roughness. This is an important quantity in printed nanosheet devices. Dielectric enhancement has been observed in printed nanosheet microcapacitors by replacing the silver nanoparticle electrodes with graphene nanosheets (Science Bulletin, **67**, 2541-2549, 2022. DOI). Here, the enhanced surface roughness led to a larger contact area and increased charge trapping, which contributed to a four-fold increase in the measured dielectric constant. Alternatively, poor quality interfaces have been shown to be detrimental the performance of vertically printed 2D supercapacitors (Energy Storage Materials, **36**: 318-325, 2021. DOI).

Most importantly, we utilise these high-resolution volumes to quantitatively measure key network properties including porosity, specific surface area, pore shape/size, nanosheet orientation, the tortuosity of the nanosheet and pore volumes, and determine the degree of nanosheet aggregation during the ink deposition process. Many of these properties are non-trivial to measure using other techniques, while some have not been reported at all. We will discuss this in detail in response to Point 2. As an example, the porosity of nanosheet networks is often qualitatively discussed from SEM cross-sections (Nature Communications, **8**, 2017. DOI), determined using sample weighing (Science, **356**: 69-73, 2017. DOI) or measured using MIP/BET measurements. Sample weighing offers no information on the nature of the pore/nanosheet networks, while the unsuitability of MIP/BET analysis for characterising printed thin-films is discussed in detail in our response to Reviewer 3 (Point 2). Finally, we present a novel study on the effect on nanosheet/nanowire size on network morphology, which we directly link to network electrical resistivity and interface quality in printed heterostacks. A means to tune the physicochemical properties of these systems by simply altering the nanosheet/nanowire dimensions, without any post-processing steps, is presented. This is made possible by the novel morphological insights into these nanoscale networks that FIB-SEM-NT can provide.

Point 1: Although this study shows some high-resolution structural images of Printed Nanostructured Networks, the FIB-nanotomography method itself is not new, as evidenced by previous publications (Catalysis Today, 405-406,1, 2022; Ultramicroscopy, 163, 38-47, 2016; Journal of Microscopy, 281: 76-86, 2020).

Response:

FIB-SEM NT has indeed been utilised to characterise the structure of a range of materials, although not printed networks and not at the resolution/sample volume reported here (see below). We have now included references to previous demonstrations of the FIB-SEM-NT technique to the main text (page 2):

“FIB-SEM nanotomography (FIB-SEM-NT) effectively bridges the gap between these tomographic techniques by offering spatial resolutions of a few nanometres over representative sample volumes of 10² – 10⁴ μm³ ³⁶. This has been demonstrated through high-resolution reconstructions of oil shales³⁷, drug release coatings³⁸, fuel cells³⁹ and commercial battery electrodes⁴⁰.”

However, it is worth noting that the works cited in Point 1 describe:

- 1) A review of the FIB-SEM tomography technique and its specific application to study porous catalyst materials (Catalysis Today, **405-406**: 2-13, 2022. DOI). These include porous solids such as zeolite crystals (Angewandte Chemie International Edition, **50**: 1294-1298, 2011. DOI), ceramics (Journal of Microscopy, 216: 84-95, 2004. DOI) and support materials for catalysts (ACS Catalysis, **11**: 4784-4798, 2021. DOI). The application of FIB-SEM NT to reconstruct small volumes (~8 μm³) of

microbead catalyst particles is also discussed (ACS Catalysis, **6**: 3158-3167, 2016. DOI) (cf. volumes of $10^2 - 10^3 \mu\text{m}^3$ in this work).

- 2) The application of FIB-SEM NT to reconstruct gold microstructures with ligament sizes on the order of tens of nanometres, prepared by corrosion of solid $\text{Au}_{30}\text{Ag}_{70}$ alloys (Ultramicroscopy, **163**: 38-47, 2016. DOI).
- 3) The challenges in imaging and segmenting soft, poorly conducting polymers, with a focus on ethyl cellulose polymer films for drug release coatings (Journal of Microscopy, **281**: 76-86, 2020. DOI) (voxel size $\sim 10 \times 10 \times 50 \text{ nm}$). Crucially, this paper outlines routes to optimise the imaging of poorly conducting films and highlights the improved segmentation performance when using machine learning classifiers. These steps have been applied in this work by tailoring the sample preparation and imaging conditions (SI, Sections 3-4, pages 14-16), while we utilise the same random forest classifier to segment our FIB-SEM NT data (Machine Learning, **45**: 5-32, 2001. DOI) (SI, Section 3, page 14).

The physical structure, sample preparation, and target application of each of the above samples is significantly different from the printed nanomaterial networks and devices we present in the main text. Furthermore, while catalyst materials and their supports, nanoporous gold structures (Applied Physics Letters, **92**: 2008. DOI) and soft polymers (Microscopy and Microanalysis, **26**: 837-845, 2020. DOI) have been studied using FIB-SEM NT, the morphology of printed nanomaterial networks and their devices has not yet been quantitatively characterised to this extent or at this resolution using any technique. *Thus, the focus of this work is not on the FIB-SEM NT method itself, but its application to characterise the morphology of printed nanosheet/nanowire/nanoparticle networks and their devices.* We demonstrate this through a systematic morphological study on the effect of nanosheet/nanowire size, which we link to network properties such as electrical resistivity, and device properties such as interfacial roughness.

Point 2: While the paper provides insights into the influence of size on morphology, porosity, and tortuosity, the authors primarily focus on the technological advancements rather than the physicochemical property investigation of the nanostructured network.

Response:

We again thank the reviewer for the points that they raise. We have used them to improve the manuscript by more clearly linking the morphological properties measured to the physicochemical attributes of solution processed 1D/2D networks and their devices. We have also clarified some of the physicochemical investigations that were performed in this work.

In the manuscript we measure critical morphological parameters including porosity, specific surface area, pore shape/size/connectivity, nanosheet orientation and aggregation, as well as the tortuosity of the pore and nanosheet volumes using reconstructed 3D volumes from a single measurement for each sample. While we directly link these attributes to the network electrical resistivity, we would contend that these properties alone represent an important physical characterisation of these networks. It is worth noting that this host of structural information is either unattainable using other techniques due to resolution/sample limitations (i.e. x-ray CT) or would require a combination of multiple measurement techniques on different representative samples. We then link these attributes to the physicochemical properties that they influence through a systematic study of nanosheet/nanowire size for printed networks and their devices.

Network porosity and pore connectivity/tortuosity are critical parameters in nanosheet networks. The amount, size, and shape of the pores in a network heavily influence nanosheet connectivity, overlap and the nature of intersheet junctions (Nature Reviews Materials, **7**: 217-234, 2021. DOI). This is extremely relevant to printed electronics applications, where intersheet junction resistances can vastly exceed the nanosheet resistance and dominate the network mobility (Nano Letters, **11**: 16-22, 2010. DOI). The role of porosity and nanosheet

packing in determining the physicochemical properties of a nanosheet network is discussed in the context of printed thin-film transistors and capacitors in the main text (page 1), as we referenced above. Further examples pertaining to the influence of porosity/pore connectivity on the physicochemical properties of the network are offered in the manuscript (main text, page 1):

“Alternatively, network porosity and pore tortuosity determine the degree of accessible nanosheet surface area for sensing or catalysis^{17,18}, as well as electrolyte infiltration and ion kinetics in battery and supercapacitor electrodes¹⁹.”

Here, we systematically measure the network porosity (as well as the pore shape and size) as a function of nanosheet dimensions (Fig. 3C-E, main text). By measuring the porosity on a slice-by-slice basis in ~15 nm increments across hundreds of cross-sections we can assess the local structural homogeneity at an unprecedented scale (Fig. 2B, main text). We demonstrate that the pore volume in these printed nanosheet/nanowire networks is > 99% percolative (or open) for the nanosheet/nanowire sizes considered. This means that there are very few inaccessible or “dead pores” that would lower the effective nanosheet surface area for electrochemical applications. A significant degree of pore connectivity has been reported for thick vacuum filtered graphene networks (> 100 μm thickness) using MIP/BET analysis (Carbon, **171**: 306-319, 2021. DOI). However, these techniques are not appropriate to characterise printed thin-film nanosheet networks (network thicknesses of ~ 0.01 – 5 μm), meaning that this has not been previously reported for these systems. We demonstrate that the porosity, pore shape and pore size of a printed nanosheet/nanowire network can be reliably tuned by changing the nanosheet dimensions (Fig. 3C-E, main text). Such morphological tailoring generally requires sample post-processing steps such as mechanical calendaring (Advanced Materials Technologies, **5**, 2020. DOI). We also show the specific surface area of a printed nanosheet network to scale with nanosheet size, whereby networks of smaller nanosheets have an increased level of exposed nanosheet surface per unit volume/mass (Fig. 4A, main text). This has been linked to improved performance in sensing, catalytic and energy storage applications (Energy & Environmental Science, **13**, 246-257, 2020. DOI). We have clarified the role of porosity and specific surface area in determining the physicochemical properties of nanosheet devices in the main text:

Main text, page 4

“The network porosity, P , determines the nature of the intersheet junctions and influences rate performance in 2D battery electrodes⁴⁸. However, due to sample and resolution limitations this is often determined from sample weighing¹⁴ or qualitatively discussed using SEM cross-sections.”

Main text, page 6

“The dependence of network morphology on l_{NS} is further reflected in the specific surface area (SSA) of the networks (Fig. 4A). This is a key parameter that describes the accessible nanosheet surface area for sensing, catalytic and energy storage applications⁶⁷.”

A quantitative measure of the connectivity of the pore and nanosheet/nanowire volumes is then provided by calculating the tortuosity factor (Figs. 2C & 4C, main text). The nanosheet network tortuosity factor influences charge transport through the film whereby charge carriers in more well-packed networks experience less tortuous paths with fewer, more conformal junctions (ACS Nano, **17**: 2912-2922, 2023. DOI). Pore tortuosity heavily influences rate performance in nanosheet-based battery electrodes (ACS Nano, **14**, 3, 3129–3140, 2020. DOI), while in gas sensing applications the porosity, pore size and tortuosity are directly linked to gas diffusion (Applied Surface Science, **427**: 215-226, 2018. DOI). To the best of the authors knowledge, the tortuosity factor of any printed nanosheet network has yet to be reported in the literature. We show that the nanosheet and pore tortuosity factors scale with nanosheet and nanowire size, whereby printed networks of smaller nanosheets/nanowires have smaller nanosheet tortuosity factors and larger pore tortuosity factors. These measurements provide novel insights into the network structure and a means to tune its physicochemical properties. Furthermore, the ratio of the in-plane and out-of-plane tortuosities is used to

highlight the preferential in-plane alignment of nanosheets/nanowires in these printed networks. This is also quantified using angular distributions of nanosheet alignment (Fig. 2F, main text) and measured as a function of nanosheet size using the Hermans orientation factor (Fig. 4E, main text) for the printed graphene networks. Such in-plane alignment has been linked to reported directional anisotropies in conductivity (Carbon, **171**: 306-319, 2021. DOI) and mass transport (ACS Nano, **14**, 3, 3129–3140, 2020. DOI) through solution processed 2D networks. Taken together, we demonstrate that printed networks of smaller LPE nanosheets are more densely packed, with smaller circular pores, higher specific surface areas, and more tortuous pore volumes when compared to networks of larger LPE nanosheets. The relevance of the pore and nanosheet tortuosity factor to the physicochemical properties of the network has been added to the manuscript.

Main text, page 5

“The nanosheet network tortuosity factor influences charge transport through the film. Charge carriers in more well-packed networks experience less tortuous paths with fewer, more conformal junctions². Pore tortuosity heavily affects rate performance in nanosheet-based battery electrodes⁵⁴, while in gas sensing applications the porosity, pore size and tortuosity are directly linked to gas diffusion⁵⁵.”

Crucially, we then directly link these morphological properties to the electrical resistivity of printed graphene networks of different nanosheet dimensions (Fig. 4F, main text). By measuring the resistivity of each size-selected network and linking it to the nanosheet size, network porosity and tortuosity, we clearly show the impact of network morphology on electrical transport in the films. Similarly, we demonstrate that the electrical specific contact resistivity scales with the nanosheet volume fraction (1 – porosity) in printed graphene networks (Fig. S37D, SI). Here, networks of smaller sheets with lower porosities exhibited considerably lower specific contact resistivities and improved charge transfer with the bottom electrode. This again demonstrates that by changing the nanosheet size the morphology, and by extension the network electrical properties, of these printed networks can be tuned.

From a printed nanosheet/nanowire device perspective the nature of the internal interfaces is a key parameter in determining device reproducibility and performance (Nature Nanotechnology, **12**: 343-350, 2017. DOI). For example, dielectric layers in printed 2D capacitors are generally required to be > 1 μm in thickness to ensure that the conductive nanosheets in the top electrode cannot penetrate the layer and cause electrical shorts (ACS Applied Electronic Materials, **2**: 3233-3241, 2020. DOI). Similarly, charge transfer between layers of different types of nanosheets in a vertical heterostack can be limited by the nature of the interface (i.e. roughness) and the inter-nanosheet junctions (Nature Electronics, **2**: 378-388, 2019. DOI). We characterise the morphology, surface topography and electrical conductivity of sprayed networks of electrochemically (EE) and liquid phase exfoliated (LPE) graphene nanosheets (Fig. 6A, main text). We then utilise the morphological learnings from the size-dependent study to directly link network morphology and device performance in printed heterostacks using a model system of printed graphene nanosheets with a silver nanoparticle (AgNP) top electrode (Fig. 6B, main text). We demonstrate that networks of smaller LPE nanosheets exhibit enhanced continuity and improved interface quality when compared to networks of larger nanosheets – owing to their reduced porosity, smaller pores, more tortuous pore volumes and reduced surface roughness. We highlight in the main text how this can improve device performance in printed TFTs and capacitors by preventing pinholes and electrical shorts, while the surface roughness can influence the interface quality (Nanoscale, **14**: 15679-15690, 2022. DOI) and cause charge trapping (Science Bulletin, **67**: 2541-2549, 2022. DOI). This again shows that the physicochemical properties of a printed LPE nanosheet network can be tuned by changing the nanosheet size, and measured using FIB-SEM-NT. We have made the following additions to the text:

Main text, page 10

“The nature of the interfaces within a printed vertical heterostack can significantly influence device reproducibility and performance⁸¹.”

Main text, page 10

“This suggests that printed networks of smaller LPE nanosheets exhibit enhanced continuity and improved interface quality owing to their reduced porosity, smaller pores, more tortuous pore volumes and decreased surface roughness. This can help mitigate charge trapping and pinhole formation, leading to improved performance in printed transistors and capacitors^{82,83}.”

Point 3: A more comprehensive analysis and discussion should and could be achieved by providing a comparison of different imaging technologies, addressing factors such as resolution limit (voxel size) and sample destruction (which is the drawback of FIB-SEM). This would help readers better understand the advantages and limitations of the employed FIB-SEM Nanotomography method in comparison to other techniques.

Response:

We thank the reviewer for this comment. We have addressed similar points regarding a resolution comparison between tomographic techniques and sample destruction in our responses to **Point 1** and **Point 2** for **Reviewer 1**.

In our response to **Point 1** for **Reviewer 1** we compare the resolution and sampling volume of FIB-SEM NT to X-ray nano CT and electron tomography in detail. Their respective limitations for the analysis of printed 0D/1D/2D networks and their devices is discussed and highlighted through two additional figures that have been added to the SI. We have also included a far more in-depth discussion of the limitations of X-ray nano CT, electron tomography, mercury intrusion porosimetry and BET N₂ analysis for these systems to the manuscript introduction. This is also detailed in the response to **Point 1** for **Reviewer 1**.

In our response to **Point 2** for **Reviewer 1** we address the destructive and invasive sample preparation that is required for X-ray nano CT and electron tomography in detail. This can include mechanical sectioning, laser cutting, FIB milling or a combination of all these techniques to produce a sample thin enough to be imaged using these techniques. We clarify that FIB-SEM NT is a destructive tomographic technique but highlight that this destruction is highly localised to 50 × 50 μm regions and preserves the macroscopic sample/device for further studies.

Additional context regarding the above points has been added to the manuscript as detailed in our responses to **Point 1** and **Point 2** for **Reviewer 1**, to help the readers better understand the advantages and limitations of FIB-SEM NT.

Reviewer 3:

Significant results were shown in the article about characterization with 3D imaging, the study of the pore sizes, orientation (size dependence) and its versatility in 3D imaging different types of 2D nanomaterials. I had some minor questions/comments. It is related to and improved from current literature. There are enough data to reproduce the work, and the methodology is sound, meeting the expectation of the field.

The work supports the claims, and I had a few comments:

We thank the reviewer for the appreciation of our work and their comments, which have helped us to improve the manuscript as detailed below.

Point 1: Write the full name term for FIB-SEM at the first abbreviation (page 1)

Response: This has been amended in the manuscript.

Main text, page 1:

“Here, we utilise focused ion beam – scanning electron microscopy (FIB-SEM) nanotomography to quantitatively characterise the morphology of nanostructured networks and their devices using nanometre-resolution 3D images.”

Point 2: For reference 22 - Do you have more examples? From this paper, the only sample preparation is removing water and impurities and packing the samples. This doesn't seem like “considerable” sample preparation for me.

Response:

We thank the reviewer for this comment and have amended the phrasing of the manuscript for clarity. For both the BET adsorption/desorption and mercury intrusion porosimetry (MIP) techniques it is crucial that the samples are extensively degassed to remove all physisorbed material that may affect the measurement. This can include lower temperature annealing steps for prolonged periods, such as 100°C for 16 hr for a V₂O₅ nanosheet electrode (Nano Energy, **39**: 151-161, 2017. DOI), or 350°C for 8 hr for electrodes based on porous carbon nanosheets (ACS Sustainable Chemistry & Engineering, **7**: 13827-13835, 2019. DOI). Annealing at elevated temperatures would irreversibly alter the morphology of printed silver nanosheet or nanowire networks, which begin to melt/sinter around 150°C. Furthermore, flexible temperature-sensitive substrates, such as PET, have glass transition temperatures of ~78°C (Journal of the Society for Information Display, **15**: 1075 – 1083, 2007. DOI). In this work, we performed FIB-SEM NT on printed silver nanosheet networks on flexible Al₂O₃-coated PET substrates, while preserving the character of the discrete nanoplatelets (Fig. 5B, main text). For MIP, the packing of the sample in the instrument cell may also cause changes to the sample structure (Cement and Concrete Research, **30**: 1517-1525, 2000. DOI). Furthermore, at the high pressures required to achieve a maximum resolution of ~3.5 nm in MIP, the printed network structure may be deformed or even damaged irreversibly, which has been reported for hardened cement pastes (Materials Characterization, **151**: 203-215, 2019. DOI).

The key issue here is centred on the sample requirements for these processes. There are limited reports of MIP analyses on nanosheet networks due to the considerable sample masses and thicknesses that are required. To perform BET and MIP analysis on solution-processed graphene networks, vacuum filtered films >100 µm in thickness were required (Carbon, **171**: 306-319, 2021. DOI). Similarly, carbon-based supercapacitor electrodes of thickness 120 µm (area 254 mm²) were needed for BET analysis (Journal of the American Chemical Society, **135**: 5921-5929, 2013. DOI). Network thicknesses in this region (> 100 µm), that were prepared via vacuum filtration due to the sample mass requirements, would be inappropriate comparators to networks for printed electronics applications. Here, devices are printed on substrates with network thicknesses

of 0.01 – 5 μm (Nature Reviews Materials, 7: 217-234, 2021. DOI). This is addressed in the main text by performing FIB-SEM NT on printed networks that range in thickness from ~ 300 nm (Fig. 6A, main text) to a few μm for printed graphene (Fig. 1-4, main text) and $\text{WS}_2/\text{AgNS}/\text{AgNW}$ networks (Fig. 5A-C, main text), as well as device heterostacks (Fig 6B-C, main text). Such printed networks would not be close to meeting the minimum mass/thickness/absolute surface area requirements for BET or MIP. We have expanded upon the limitations of MIP/BET analysis for characterising printed thin-film networks in the main text with additional citations.

Main text, page 2

“Standard techniques such as mercury intrusion porosimetry (MIP) and N_2 BET analysis have been used to determine the pore-size-distribution and specific surface area in thick, vacuum filtered nanosheet networks²⁰. However, such methods generally require sample volumes (film thicknesses $> 100 \mu\text{m}^{20}$) that are far beyond the scope of printed thin-film electronics. Furthermore, these techniques can require high-temperature annealing pre-treatment steps²¹ that are incompatible with temperature-sensitive flexible substrates²², while sample preparation²³ and high-pressure mercury intrusion can irreversibly alter the network structure²⁵.”

Point 3: Are there limitations for the Dragonfly for the image stack alignment and 3D reconstruction? (page 3 and SI)

Response:

Dragonfly (Object Research Systems, ORS) is a 3D data visualization and analysis software that has been applied extensively across materials science (ACS Applied Nano Materials, 4: 621-632, 2021. DOI), life sciences (Scientific Data, 7, 2020. DOI), geosciences (Journal of Paleontology, 96: 152-163, 2021. DOI) and within manufacturing to reconstruct x-ray CT, FIB-SEM NT and ptychographic tomography data in 3D. We have added these applications to the alignment section of the supplementary information (Section 4, page 14).

As mentioned in the SI, the sum of squared differences (SSD) matching process was used to align adjacent slices and a linear drift compensator was employed to correct systematic drifts using the *Dragonfly* software package (SI, page 14). To optimise the alignment process several pre-processing steps were applied. The brightness and contrast of the images in each stack was first normalised to clearly distinguish between nanosheets and pores. This served to reduce the “shine-through” effect and ensured that only nanosheets at the cross-section face were present in the 2D image. Furthermore, any network cross-sections that had large translations with respect to the rest of the image stack were first manually transformed to roughly align the stack before applying the registration algorithm. Within *Dragonfly* there are several different algorithms that can be used to align an image stack (enhanced correlation coefficient, feature based, mutual information, SSD and template matching). We tested each algorithm on a network volume, finding the resultant network properties (porosity, tortuosity, and specific surface area) to be in broad agreement. However, the best performance in terms of computational cost was achieved using the SSD matching process, so this algorithm was used. A comparative figure showing the performance of the different matching algorithms has been added to the SI (Fig S13, page 14).

Fig S13: Image stack alignment for silver nanosheets (AgNS) in *Dragonfly*. (A) The original, unaligned image stack, viewed in the yz -plane. Images were captured in the xy -plane. (B-F) The same image stack aligned using the (B) Sum of squared differences (SSD), (C) Scale-invariant feature transform (SIFT), (D) Correlation Coefficient, (E) Cross-correlation and (F) Squared difference matching algorithms. (G) Measured network porosity, tortuosity (in the xy -direction) and specific surface area (SSA) for each alignment method.

SSD image matching is based on pixel-by-pixel intensity differences between the two images (IEEE Conference on Research and Development, 100-104, 2015. DOI). In terms of limitations of the SSD matching process, this algorithm can be sensitive to outliers in pixel intensity (IEEE Transactions on Pattern Analysis and Machine Intelligence, 24: 853-857, 2002. DOI). However, by applying the pre-processing steps to optimise the image brightness/contrast, this issue can be mitigated. Furthermore, the slice thickness of ~ 15 nm employed in this work means that even the smallest nanosheets ($l_{NS} = 80$ nm) were imaged >5 times. This helps the slice matching process by ensuring the difference between adjacent slices is as small as possible. Finally, as this is a pixel intensity matching process the true pixel size is irrelevant to the SSD algorithm, meaning the spatial resolution is limited only by the resolution that the images were experimentally captured at.

Point 4: Does the spray control the placement/distribution of graphene? If so, how would that affect the diffusive flux (page 4)

Response:

We thank the reviewer for this insightful comment. Spray parameters are indeed known to affect morphology and overall network uniformity, including the placement and distribution of deposited nanomaterials (although this has never been properly quantified till now). This is something that we have attempted to optimise and control for in this work. Each network was deposited using a computer controlled xyz -gantry to define the

spray area. Within this region, shadow masks were used to deposit highly-defined sample traces. We have added an image of a patterned nanosheet network trace to the SI to highlight this (Fig. S3B).

Figure S3: UV-vis of LPE graphene inks and deposition into a network trace. (A) Optical extinction spectra for the size-selected graphene inks normalised to the dimension independent plateau at 750 nm. (B) Spray-cast network of graphene nanosheets patterned onto a glass substrate using a shadow-mask. Evaporated gold bottom electrodes were used for electrical measurements.

Regarding the influence of spraying parameters on network uniformity and nanostructure, parameters include the airbrush stand-off distance and raster speed, the N_2 gas back pressure and ink flow rate, as well as the substrate temperature and nozzle diameter. Both the ink droplet size and drying rate influence network structure through drying and material aggregation effects. Through appropriate wetting, a film can flow to fill voids and reduce porosity, while deposition of dry particles can lead more granular and aggregated structures. (Flexible and Printed Electronics, **3**: 035002, 2018. DOI) This can affect the pore/material tortuosity, network porosity, nanosheet alignment and the distribution of intersheet junctions. Previous work on sprayed carbon nanotube networks determined that elevated temperatures are critical to suppress drop coalescence and coffee-stain formation on the substrate (Chemical Engineering Science, **65**: 2000-2008, 2010. DOI). Multivariate analysis on sprayed silver nanowire networks found that the parameter with the most influence on network uniformity and material aggregation was the N_2 backpressure used to aerosolise the ink (Small, **7**: 2621-2628, 2011. DOI). An N_2 backpressure of ~ 45 psi was found to produce a more homogeneous distribution of smaller ink droplets, facilitating expedited solvent evaporation for improved network uniformity and opto-electrical properties. As we show in the main text (Fig. 2C, Fig. 4C and Fig. 5D-F) nanosheet/nanowire networks that are aligned in the plane of the film will demonstrate much larger tortuosity factors in the out-of-plane direction. Aggregated and disordered networks demonstrate more similar tortuosity factors in the in-plane and out-of-plane directions. To ensure consistent deposition across the samples sprayed in this work, the spraying parameters were fixed to the optimised conditions outlined in (Small, **7**: 2621-2628, 2011. DOI). Additional details and citations have been added to the *Network Deposition* section of the manuscript (main text, page 14).

“Nanomaterial inks were spray coated using a Harder and Steenbeck Infinity Airbrush attached to a computer-controlled Janome JR2300N mobile gantry. A N_2 back pressure of 45 psi, nozzle diameter of 0.4 mm and stand-off distance of 100 mm between the nozzle and substrate were used⁹⁰. All traces were patterned using stainless-steel masks while the substrate was heated to 80°C using a hotplate. The size-selected LPE graphene inks were sprayed at a concentration of 0.2 mg.ml⁻¹ onto ultrasonically cleaned glass slides with prepatterned gold electrodes (Temescal FC2000 metal evaporation system) to facilitate electrical measurements.

Point 5: Can the spray also affect the porosity study shown in Fig 3?

Response:

By changing the spray parameters the porosity of the resultant network can be affected, as we describe in our response to **Point 4**. To ensure that we isolated only the effect of changing nanosheet/nanowire dimensions in the main text (Figs. 3 & 5), we fixed the spraying parameters for each size selected ink using optimised values in-line with (Small, 7: 2621-2628, 2011. DOI). All inks in the porosity study were diluted to the same concentration for each material and deposited over the same area using a computer-controlled xyz-gantry at a stand-off distance of 100 mm. For each comparison the substrate, platen temperature, nozzle diameter (0.4 mm) and N₂ back pressure (45 psi) was kept consistent. This is described in detail in the *Network Deposition* section of the manuscript (main text, page 13).

We note that the ability to use FIB-SEM NT to quantify morphology would allow one to quantify the effects of deposition parameters on network structure for the first time. This we see as future work.

Point 6: Figure 4D - it is hard to see Figure 4D

Response: The font size on all titles, legends, and labels in Fig. 4D has been increased.

REVIEWERS' COMMENTS

Reviewer #1 (Remarks to the Author):

The authors have addressed the issues well. This version is fine to be accepted.

Reviewer #3 (Remarks to the Author):

This great work shows the characterization with 3D imaging, pore sizes and orientation. There is enough data to reproduce the work, and the methodology is sound, meeting the expectation of the field. I am satisfied with their responses and explanations. The responses added to their manuscript and SI made it sufficient. I would recommend the revised manuscript be accepted.